
# The value of ASCAT soil moisture and MODIS snow cover data for calibrating a conceptual hydrologic model

Rui Tong[1,2], Juraj Parajka[1,2], Andreas Salentinig[3], Isabella Pfeil[1,3], Jürgen Komma[2], Borbála Széles[1,2], Martin Kubáň[4], Peter Valent[2,4], Mariette Vreugdenhil[3], Wolfgang Wagner[1,3] and Günter Blöschl[1,2]

[1]Centre for Water Resource Systems, TU Wien, Vienna, 1040, Austria
[2]Institute of Hydraulic Engineering and Water Resources Management, TU Wien, Vienna, 1040, Austria
[3]Department of Geodesy and Geoinformation, TU Wien, Vienna, 1040, Austria
[4]Department of Land and Water Resources Management, Slovak University of Technology, Bratislava, 810 05, Slovakia

*Correspondence to*: Rui Tong (tong@hydro.tuwien.ac.at)

**Abstract.** Recent advances in soil moisture remote sensing have produced satellite datasets with improved soil moisture mapping under vegetation and with higher spatial and temporal resolutions. In this study, we evaluate the potential of a new, experimental version of the ASCAT Soil Water Index dataset for multiple objective calibration of a conceptual hydrologic model. The analysis is performed in 213 catchments in Austria for the period 2000-2014. An HBV type hydrologic model is calibrated to runoff data, ASCAT soil moisture data, and MODIS snow cover data for various calibration variants. Results show that the inclusion of soil moisture data in the calibration mainly improves the soil moisture simulations; the inclusion of snow data mainly improves the snow simulations; and including both of them improves both soil moisture and snow simulations to almost the same extent. The snow data are more efficient in improving snow simulations than the soil moisture data are in improving soil moisture simulations. The improvements of both runoff and soil moisture model efficiencies are larger in low elevation and agricultural catchments than in others. The calibrated snow-related parameters are strongly affected by including snow data, and to a lesser extent by soil moisture data, while the soil-related parameters are only affected by the inclusion of soil moisture data.

# 1 Introduction

Estimating the spatial and temporal variability of water balance components at the regional scale is important for solving a range of practical issues in water resources management and planning, as well as for understanding catchment functioning in terms of how runoff generation processes interact to produce catchment response. One estimation approach is by hydrologic models. There is a variety of model types and model parameters estimation methods. Notwithstanding their usefulness, the resulting simulations of the water balance components are subject to uncertainty due to uncertainty in model inputs, parameter estimation and model structure (Parajka et al., 2007).





Previous studies have demonstrated that multiple objective calibration helps to constrain hydrologic models and hence to reduce uncertainty and to improve predictions in hydrological modelling (e.g. Efstratiadis and Koutsoyiannis, 2010). Most of these studies examined the value of constraining hydrologic models by combining different runoff signatures (e.g. by simultaneous calibration of the models to low and high flows or timing) or calibrating hydrologic models to runoff and some additional hydrological variable such as snow cover (e.g. Udnæs et al., 2007; Parajka and Blöschl, 2008; Franz and Karsten,

2013; Duethmann et al., 2014; Finger et al., 2015, Han et al., 2019, Sleziak et al., 2020), soil moisture (e.g. Parajka et al., 2009; Sutanudjaja et al., 2014; Wanders et al., 2014; Rajib et al., 2016; Kundu et al., 2017,  Li et al., 2018), evaporation (e.g. Immerzeel and Droogers, 2008; Zhang et al., 2009), groundwater level data (Seibert, 2000) or total water storage (e.g. Lo et al., 2010; Werth and Güntner, 2010; Rakovec et al., 2016; Bai et al., 2018; Trautmann et al., 2018). These studies showed that the use of additional information typically improves the spatial and/or temporal patterns of internal state variables and fluxes,

but does not necessarily improve the efficiency of simulating runoff. Most of the studies reported a small degradation of runoff model efficiency while internal consistency of the models has been improved. A few studies also tested the combination of more variables in multiple objective calibration (e.g. Milzow et al., 2011; Kunnath-Poovakka et al., 2016;  López et al., 2017; Nijzink et al., 2018; Demirel et al., 2019; Szeles et al., 2020a,b) and found that the combination of different variables generally reduced the parameter uncertainty particularly in data poor regions. Nijzink et al (2018) also demonstrated that constraining

hydrologic models profited from an increased number of data sources. Interestingly, combining different soil moisture and satellite products had a positive impact on the identifiability of not only soil but also snow model parameters.

The factors that control these improvements are less well understood. Snow cover data improved snow and runoff simulations in small catchments without precipitation observations (Parajka and Blöschl, 2008) and helped to reduce snow underestimation errors in flatland catchments in changing climate conditions (Sleziak et al., 2020). Including evapotranspiration estimates

improved regionalization and simulations of daily and monthly runoff particularly in drier regions with lower runoff volumes. (Zhang et al., 2009). The use of total water storage data from GRACE improved runoff simulations on monthly or longer time scales, particularly in wet catchments (Rakovec et al. 2016). Only a few studies examined the factors that control the changes in efficiency when using soil moisture information (Rajib, et al., 2016). Parajka et al., (2006) showed that soil moisture efficiency of a model calibrated to runoff and satellite soil moisture was lower in hilly and alpine regions with large

topographical variability as compared to the flatlands. Nijzink et al. (2018) reported that the AMSR-E soil moisture product improved the identifiability of model parameters in peaty lowland catchments.

Recent advances in the observation techniques of soil moisture, particularly in passive and active microwave remote sensing, increases the availability of regional and global soil moisture datasets (Babaeian et al., 2019). Passive microwave sensors operating in the 1-10 GHz frequency range suitable for soil moisture retrieval include the L-band radiometers flown on board

the Soil Moisture Active Passive (SMAP) and Soil Moisture and Ocean Salinity (SMOS) missions, and the multi-frequency radiometer Advanced Microwave Scanning Radiometer 2 (AMSR-2). In the active domain, soil moisture retrievals from the C-band Advanced Scatterometer (ASCAT) have found widespread use in geoscientific applications (Brocca et al. 2017). All these satellites have a rather coarse spatial resolution in the order of tens of kilometres (10-50 km). Various validation studies





have shown that ASCAT soil moisture data sets (Wagner et al. 2013) are less accurate than corresponding SMAP soil moisture
data sets (Kim et al. 2020) but, overall, comparable in quality with SMOS and AMSR-2 (Chen et al., 2018; El Hajj et al., 2018;
Mousa and Shu 2020). Nonetheless, there are important regional differences in the quality of the satellite soil moisture data
sets, with ASCAT performing in general poorest over arid environments and best over more densely vegetated regions. Over
the United States and Europe, comparisons with in-situ soil moisture data from dense networks have revealed the presence of
seasonal biases in the ASCAT soil moisture time series (Wagner et al., 2014). Pfeil et al. (2018) and Hahn et al. (2020) have
demonstrated that these seasonal biases can be much reduced by enhancing the vegetation parameterization of the TU Wien
change detection model introduced by Wagner et al. (1999). The launch of the Sentinel-1 series provides observations at a
high spatial resolution of 5x20 m. Over mountainous environments soil moisture retrievals from all microwave sensors are in
general much less reliable than over flatland regions due to significant topographic variations within the coarse-resolution
satellite footprints and the presence of rocks, ice, snow, and dense vegetation. Nonetheless, in the snow- and frost-free summer
months, the satellite retrievals may have some skill as demonstrated by Brocca et al. (2013) for an alpine catchment in northern
Italy.

The objective of this study is to test the value of a new ASCAT Soil Water Index (SWI) data product for multiple objective
calibration and validation of a conceptual hydrologic model. Compared to the operational ASCAT SWI product as distributed
by the Copernicus Global Land Service, this experimental SWI data product mainly benefits from a new vegetation
parameterization of the ASCAT surface soil moisture retrieval algorithm and an improved spatial representation due to the
application of a new directional resampling method based on Sentinel-1 Synthetic Aperture Radar (SAR) data. The main aims
are: (1) to evaluate the performance of a conceptual hydrologic model calibrated to satellite soil moisture and runoff and to
test the impact of weight on the runoff objective in model calibration, (2) to compare the multiple objective calibration to soil
moisture and runoff to three different calibration variants – (i) traditional calibration to runoff only, (ii) multiple objective
calibration to satellite snow cover and runoff and (iii) multiple objective calibration to satellite snow cover, soil moisture and
runoff; (3) to examine factors which control the model performance at the regional scale. The analysis is performed by
confronting a conceptual hydrologic model with ASCAT SWI soil moisture data for 213 catchments in Austria, which
represent a wide range of physiographic settings typical of Central European conditions.

# 2 Data

## 2.1 ASCAT Soil Water Index product

For producing a new, experimental version of the ASCAT Soil Water Index (SWI) data set we deployed the same algorithms
as used within the EUMETSAT H SAF and Copernicus Global Land Service, but with a new parameterization for the
vegetation correction (Hahn et al., 2020) and a new approach for disaggregating the ASCAT soil moisture retrievals to a finer
grid, which is currently under review by H SAF for producing the planned Metop ASCAT Disaggregated Surface Soil Moisture





Near Real Time 1 km sampling (ASCAT DIS SSM NRT 1 km - H28) data product. The main steps in the processing are: (i) retrieving surface soil moisture from ASCAT backscatter time series using the TU Wien change detection algorithm adopted to ASCAT (Naeimi et al., 2009) and using the vegetation parameterization as recommended by Pfeil et al. (2018), (ii) disaggregating the ASCAT surface soil moisture data to a regular 500 m grid using the directional resampling method as described in the Algorithm Theoretical Baseline Document (ATBD) for the planned H SAF H28 data product, and (iii)

computing the Soil Water Index (SWI) using the iterative implementation of the exponential filter introduced by Wagner et al. (1999) and Albergel et al. (2008) with a characteristic time value of T = 10 days, representing root zone soil moisture. This last processing step makes the ASCAT soil moisture data better comparable to the modelled soil moisture data as it filters out high-frequency fluctuations of the ASCAT surface soil moisture retrievals and samples the data at regular time intervals. The disaggregation step is based on the analysis of Sentinel 1 backscatter time series sampled to 500 m, and essentially looks for

the best direction from which the ASCAT data are interpolated to the 500 m grid. Thereby it improves the resampling especially near large lakes or near large urban areas. To exclude invalid ASCAT measurements of snow and frozen ground, soil moisture was masked using soil temperature and snow cover from ECMWF Copernicus Climate Service (C3S) ERA5-Land. Soil moisture was masked when soil temperatures at a soil depth of 0-7 cm were below 1°C or snow cover exceeded 30 % of the pixel.

## 2.2 MODIS snow cover product

Snow cover is mapped by combining the MODIS products from the Terra (MOD10A1) and Aqua (MYD10A1) satellites (Hall and Riggs, 2016a, b). Version 6 of the MOD10A1 and MYD10A1 datasets provides daily maps of Normalized Difference Snow Index (NDSI) at a 500 m spatial resolution. The NDSI values range between 0.0 and 1.0 and snow cover is considered to be present if NDSI is larger than a threshold. Former MODIS versions used a fixed threshold (0.4), but Tong et al. (2019)

found that in Austria this threshold can be seasonally optimized for different altitudes and land cover classes. In this study we use a threshold that varies seasonally, decreases with increasing elevation and is lower in forested than open land cover settings (Table 1). Such a varying threshold improves the regional snow cover mapping by 3-10%, mainly in forested regions above 900m a.s.l. (Tong et al., 2019). The classified snow cover maps from Terra and Aqua are then combined to reduce the effect of clouds. Pixels classified as clouds or missing in Terra are replaced by pixels from Aqua if these are classified as snow

covered or snow free (Parajka and Blöschl, 2008).

## 2.3 Study area and other data

The value of satellite data for calibration of hydrologic models is evaluated in 213 catchments in Austria (Figure 1, Table 2). This set of catchments has been selected in previous studies (Viglione et al., 2013, Sleziak et al., 2020) to represent diverse physiographic, landscape and hydrologic characteristics which are not significantly affected by human impact. Selected

catchment characteristics of this dataset are presented in Table 2. The size of the catchments varies from 13.7 to 6214 km$^2$ and





their mean elevation ranges from 353 m a.s.l. to 2940 m a.s.l.. Topographical characteristics are derived from digital elevation model with 500m spatial resolution. Land cover in Austria is mainly agricultural crops and meadows in the lowlands and forest in the medium elevation ranges. Alpine vegetation and rocks prevail in catchments in the Alps. Land cover characteristics are derived from the CORINE land cover mapping (CLC2006 dataset, EEA, 2013, https://land.copernicus.eu/) and Normalized

Difference Vegetation Index (MOD13A3v006) is generated from MODIS C6 1km monthly data (Didan, 2015). Austria has a warm temperate climate, except of the Alps. The largest precipitation rates (more than 2000 mm/year) occur in the west, mainly due to orographic lifting of northwesterly airflows at the rim of the Alps. Mean annual catchment precipitation is lower (less than 800mm/year) in the lowlands in the east, and the contrast with the Alps is reinforced by the higher air temperature and much higher evaporation in the lowlands. Soil characteristics are derived from a 1km global map of soil hydraulic properties

(Zhang et al., 2018). This dataset provides the mean and standard deviations of selected soil hydraulic parameters based on the Kosugi water retention model (Kosugi 1994, 1996) at a 1 km resolution for surface soils (0-5cm).

Hydrological and meteorological data are obtained from Central Hydrographical Bureau (HZB, ehyd.gv.at) and Zentralanstalt für Meteorologie und Geodynamik (ZAMG). Model inputs (i.e. mean daily precipitation and air temperature) are derived from the gridded SPARTACUS dataset (Hiebl and Frei, 2016, 2018). This dataset provides long-term daily gridded (1km spatial

resolution) maps which are consistently interpolated by using a consistent station network throughout the entire period (Duethmann et al., 2020). Mean daily potential evaporation is derived from gridded maps of mean daily air temperature and potential sunshine duration index by using a modified Blaney–Criddle approach (Parajka et al., 2003). Daily runoff data from 213 catchments are used for calibrating and 208 catchments for validating the hydrologic model.

The precipitation, air temperature, runoff and MODIS snow cover data are available from September 2000 to August 2014.

The concurrent available period for the soil moisture ASCAT SWI data is January 2007 to August 2014.

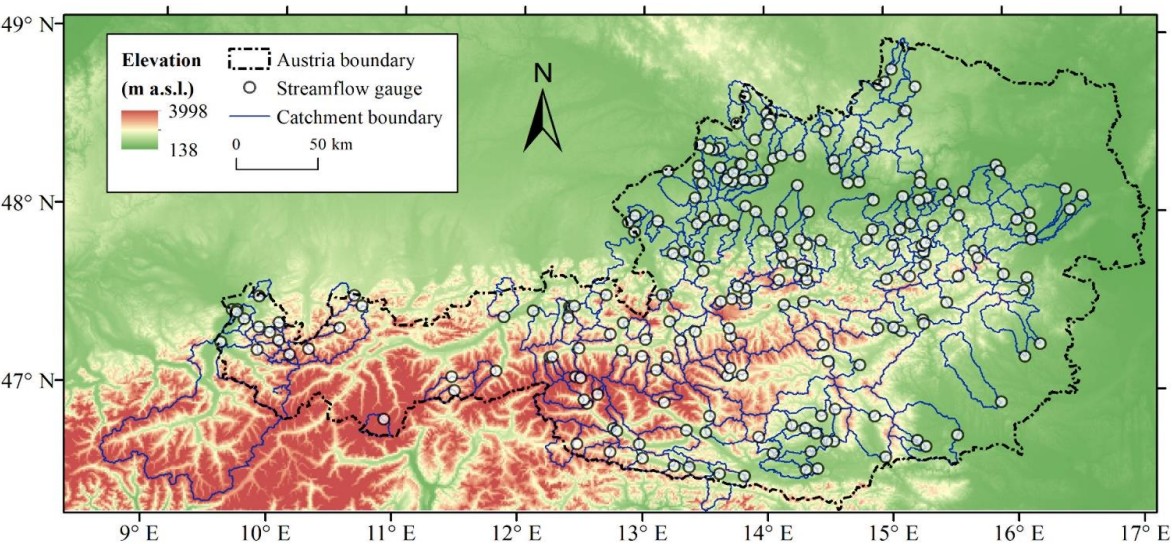

**Figure 1: Topography of Austria and the location of the 213 catchments.**



# 3 Methods

## 3.1 Conceptual hydrologic model

The hydrologic model used in this study is a semi-distributed version of TUW model, following the structure of the HBV model (Bergström 1992; Parajka et al., 2007). A simple illustration of the model structure is presented in Figure 2. The model consists of three routines, i.e. snow accumulation and melt, soil moisture accounting and runoff routine. The snow part has five model parameters and is based on a simple degree-day method: the snow correction factor SCF to account for errors in measurement of snowfall due to gauge undercatch, the degree day factor DDF, and three threshold temperatures Ts, Tr, meltT.

The soil moisture routine has three parameters: the maximum soil moisture storage in the root zone FC, a limit that controls the actual evapotranspiration LP and a nonlinear parameter for runoff production beta. The routing involves two parts, within-catchment routing and stream routing. The within-catchment routing has five parameters: three storage coefficient k0, k1 and k2 for three conceptual reservoirs representing overland flow, interflow and base flow, a threshold for very fast response lsuz and a constant percolation rate cperc connecting the fast and slow reservoirs. The stream routing uses a triangular transfer

function with two parameters: bmax and croute. The total number of model parameters that are calibrated is 15 (Table 3). The model is run in a semi-distributed way, i.e. model inputs and outputs are estimated for elevation zones of 200m while the model parameters are assumed to be lumped (i.e. constant) in each catchment. In order to match the model simulations to the dimensionless satellite soil water index, the simulated soil moisture is scaled by the field capacity (i.e. the  model parameter FC), to obtain a relative root zone moisture ranging from 0 to 1. More details about the model can be found in the Appendix

of Parajka et al. (2007).

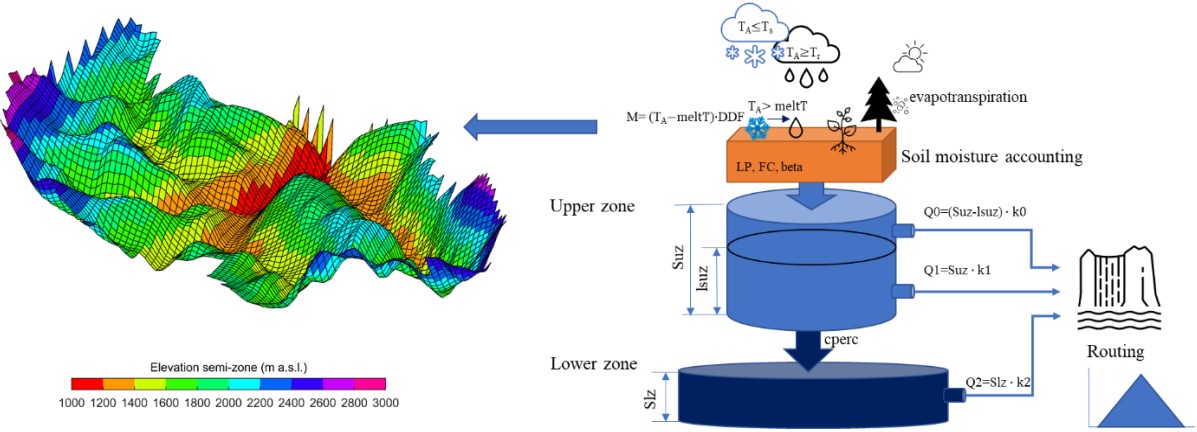

**Figure 2: Conceptual description of TUWmodel structure.**





## 3.2 Multiple objective calibration and validation of hydrologic model

The value of using satellite soil moisture (SSM) data for multiple objective calibration of conceptual hydrologic models is

compared to three other calibration variants: (1) traditional calibration to runoff only, (2) multiple objective calibration to satellite snow cover (SSC) and runoff and (3) multiple objective calibration to SSM, SSC and runoff. The general form of the calibration objective function *F* used in this study consists of minimizing the weighted sum of individual objectives related to runoff ($O_Q$), soil moisture ($O_{SM}$) and snow cover ($O_{SC}$):

$$F = w_Q \cdot O_Q + w_{SM} \cdot O_{SM} + w_{SC} \cdot O_{SC} \qquad (1)$$

where $w_Q$, $w_{SM}$ and $w_{SC}$ are the weights of the respective objectives. In each multiple objective calibration variant, 11 runoff weights (from 0.0 to 1.0 with a step of 0.1) are tested (Table 4). The soil moisture and snow weights are assumed to be equal for symmetry, and are calculated by setting the sum of all weights to 1.0.

The individual objectives $O_Q$, $O_{SM}$ and $O_{SC}$ are defined as follows. The runoff objective $O_Q$ consists of a combination of two variants of the Nash-Sutcliffe coefficient *NSE* and *NSE$_{log}$* (Nash and Sutcliffe, 1970):

$$O_Q = 0.5 \cdot NSE + 0.5 \cdot NSE_{log} \qquad (2)$$

$$NSE = 1 - \frac{\sum_{i=1}^{n}\left(Q_{obs,i} - Q_{sim,i}\right)^2}{\sum_{i=1}^{n}\left(Q_{obs,i} - \overline{Q_{obs}}\right)^2} \qquad (3)$$

$$NSE_{log} = 1 - \frac{\sum_{i=1}^{n}\left(\log(Q_{obs,i}) - \log(Q_{sim,i})\right)^2}{\sum_{i=1}^{n}\left(\log(Q_{obs,i}) - \log(\overline{Q_{obs}})\right)^2} \qquad (4)$$

where $Q_{obs,i}$ and $Q_{sim,i}$ represent observed and simulated daily runoff of day *i*, respectively and $Q_{obs}$ is the average of observed daily runoff over the calibration (or verification) period of n days. The choice of equal weighting of *NSE* and *NSE$_{log}$* is based

on previous studies in the study region (e.g. Parajka and Blöschl, 2008) to emphasize both the high and low flow conditions. The soil moisture objective function ($O_{SM}$) is expressed by the correlation coefficient *r* between relative soil moisture estimated from the ASCAT and simulated by the hydrologic model:

$$O_{SM} = \frac{\sum_{i=1}^{n}\left((\theta_{sim,i} - \overline{\theta_{sim}})(\theta_{obs,i} - \overline{\theta_{obs}})\right)}{\sqrt{\sum_{i=1}^{n}\left((\theta_{sim,i} - \overline{\theta_{sim}})^2(\theta_{obs,i} - \overline{\theta_{obs}})^2\right)}} \qquad (5)$$





where $\theta_{sim}$ is the relative root zone soil moisture simulated by the model and $\theta_{obs}$ is the ASCAT SWI. The correlation

coefficient is selected as a measure of similarity because it allows a comparison of the temporal dynamics irrespective of the

respective magnitudes and possible intercepts in the relationship between observed and simulated soil moisture.

The snow cover objective function $O_{SC}$ involves the sum of snow overestimation $S_O$ and underestimation $S_U$ errors:

$$O_{SC} = 1 - (S_O + S_U) \qquad (6)$$

The estimation of $S_O$ and $S_U$ follows the strategy proposed and evaluated in Parajka and Blöschl (2008). The snow

overestimation error indicates the relative number of days when the hydrologic model simulates snow but the satellite (MODIS)

does not observe snow cover:

$$S_O = \frac{1}{\sum\limits_{i=1}^{N_{days}} \sum\limits_{j=1}^{N_{zones}} A_{i,j}} \sum\limits_{i=1}^{N_{days}} \sum\limits_{j=1}^{N_{zones}} A_{i,j} \bigcap (SWE_{i,j} > \xi_{SWE}) \bigcap (SCA_{i,j} = 0) \qquad (7)$$

where $A_{i,j}$ is the area of zone $j$ on day $i$ which is cloud free from MODIS. $SWE_{i,j}$ is the simulated snow water equivalent in

elevation zone $j$ larger than 10mm, $SCA_{i,j}$ is the snow covered area estimated from MODIS within this zone, $N_{days}$ is the number

of days i, where MODIS images are available with cloud cover less than a threshold ($\xi_C$) 50%.

The snow underestimation error indicates the relative number of days when the hydrologic model does not simulate snow in a

zone, but MODIS indicates that snow covered area greater than a threshold 25% is present in the zone, i.e.:

$$S_U = \frac{1}{\sum\limits_{i=1}^{N_{days}} \sum\limits_{j=1}^{N_{zones}} A_{i,j}} \sum\limits_{i=1}^{N_{days}} \sum\limits_{j=1}^{N_{zones}} A_{i,j} \bigcap (SWE_{i,j} = 0) \bigcap (SCA_{i,j} > \xi_{SCA}) \qquad (8)$$


The snow covered area, *SCA*, within each zone is calculated from the MODIS data as

SCA=S/(S+L)    (9)

where S and L represent the number of pixels mapped as snow and snow free, respectively, for a given day and a given

elevation zone.

The thresholds $\xi_{SWE}$, $\xi_{SCA}$ and $\xi_C$ are chosen on the basis of the sensitivity analysis performed by Parajka and Blöschl (2008).

All calibration variants are automatically calibrated by using the shuffled complex-self adaptive hybrid evolution (SC-SAHEL)

developed by Naeini et al. (2018). It combines four evolutionary algorithms (EA) with self-selected scheme and hence the

evolution process of generating parameter values is more robust. The number of complexes is set to 8, allowing the four EAs

to be automatically changed by each evolution generation. The optimization is stopped at any of the following three criteria:

if the parameters converge to a space of geometric size less than 0.01; if the best objective function value has not improved by

0.1% over the last 10 loops; if the total number of runs reaches 1,000,000 (see. Chu et al., 2011; Naeini et al., 2018).





The calibration period used in all variants is from September 1, 2000 to August 31, 2010. The validation period is from September 1, 2010 to August 31, 2014. The warmup period is one year before the start of the calibration or validation period. Since soil moisture satellite data are available only from January 2007, the soil moisture simulation efficiency for the calibration period is calculated for a shorter time period.

# 4 Results

## 4.1 Performance of multiple objective calibration

The calibration model performance of three multiple objective calibration variants is presented in Figure 3 and Table 5. The objective function involves a runoff component weighted by $w_Q$ and additional soil moisture and snow components (Table 4). The limiting case is $w_Q$=1 where only the runoff component is used and this case represents a typical calibration to runoff only. The case where $w_Q$=0 represents calibration only to SSM and/or SSC without the use of runoff data. The median runoff efficiency over the 213 catchments (Figure 3, top; Table 5) ranges between 0.74 and 0.79 for $w_Q$ larger than 0.3 irrespective of the calibration variants. Using SSM and SSC with runoff for model calibration results in a similar pattern of model performance to the case when only SSM and runoff are used, but the former variant has a smaller regional variability (scatter) of runoff model efficiency for $w_Q$ less than 0.6. Interestingly, when no runoff is involved ($w_Q$=0), using only SSC results in slightly better runoff simulations than when using only SSM, while for $w_Q$ from 0.1 to 0.3 the opposite is the case.

The correlation between ASCAT and simulated soil moisture (Figure 3, centre) has a much larger regional variability (i.e. variability between catchments) than the $w_Q$ variants. For the SSM and runoff variant, the median of correlation increases from 0.29 to 0.52 with decreasing $w_Q$, and the variant using all three variables is similar. For the snow cover and runoff variant $w_Q$ has little effect on soil moisture correlation and correlation is similar to the runoff only calibration ($w_Q$=1).

Similar patterns are observed for the snow cover efficiency. The SSM weighting has little effect on the snow cover simulations. The median $O_{SC}$ is between 0.75 and 0.79 for $w_Q$ larger than 0.0. The variants that use SSC show increasing performance with decreasing $w_Q$ and the regional variability decreases. For $w_Q$ less than 0.5, the median $O_{SC}$ is between 0.84 to 0.91, which is 5 to 13% larger than the median for calibration to runoff only ($O_{SC}$ =0.79). These results indicate that the simultaneous use of SSM and SSC in model calibration can improve simulations of soil moisture and snow cover in the calibration period, without any significant reduction in runoff model efficiency, particularly for $w_Q$ between 0.3 and 0.4.





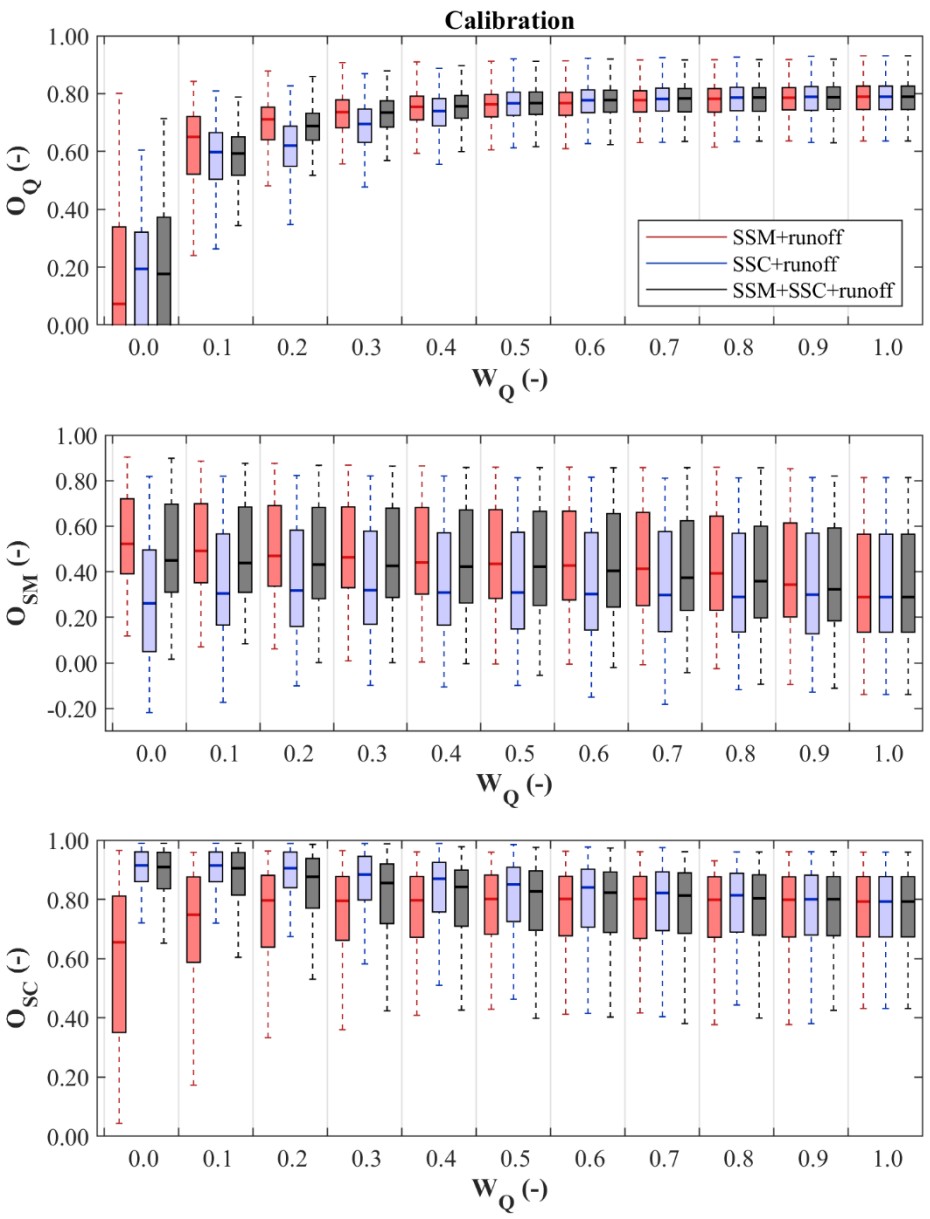

**Figure 3: Hydrologic model performance for three multiple objective calibration variants: calibration to satellite soil moisture and runoff (red boxes), to satellite snow cover and runoff (blue boxes) and to satellite soil moisture, snow cover and runoff (grey boxes). Top, middle and bottom panels show runoff (Eq. 2), soil moisture (Eq. 5) and snow cover (Eq. 6) model efficiency for different weights of the runoff objective $w_Q$ in the calibration period 2000-2010, respectively. $w_Q =1$ represents calibration to runoff only. Size of the boxes represents the spatial variability across the 213 catchments.**


The model performance for the validation period (2010-2014) is presented in Figure 4 and Table 6. The patterns of changing

model efficiency with changing $w_Q$ are very similar to those in the calibration period. The median of validation runoff model

efficiency of the SSM and runoff calibration variant for $w_Q > 0.3$ is between 0.71 and 0.73, which is similar or only somewhat

smaller than that for calibration to runoff only ($w_Q = 1$). The SSC and runoff calibration variant results show a slightly lower

runoff model performance for weights $w_Q < 0.3$. The calibration with all three variables gives practically identical validation

efficiencies as the variant with SSM and runoff.

The median soil moisture correlation increases from 0.43 to 0.54 with decreasing $w_Q$ for the SSM and runoff calibration variant

and ranges from 0.42 to 0.49 for the variant that uses all variables. The smallest correlations are found for the SSC and runoff

variant, where median of correlation $r$ varies between 0.35 and 0.43. The regional variability in $r$ is however much larger for

all variants than for the calibration period. The scatter (i.e. difference between 75- and 25- percentiles) in $r$ is around 0.3 for

all $w_Q$. For the variants that include SSM the 75-percentiles vary between 0.60 and 0.68.

The snow cover efficiency for $w_Q$ larger than 0.5 is very similar for all three variants. For $w_Q$ smaller than 0.5, $O_{SC}$ tends to

increase and the regional variability decreases for the variants involving SSC. The validation $O_{SC}$ is about 2% larger than that

obtained in the calibration period. Similar to the calibration period, the weighting of SSM and runoff has hardly any impact on

$O_{SC}$. Adding SSC data to SSM and runoff improves the snow simulation, particularly for $w_Q$ less than 0.4.

It is also interesting to compare the relative performances in the validation period to that in the calibration period.

The runoff model performance always decreases when moving from the calibration to the validation period, although the

decrease is relatively small, suggesting that there is no overfitting. The soil moisture model performance in contrast always

increases when moving from the calibration to the validation period. This is likely because in the case of soil moisture the

calibration period only consists of about four years. The snow model performance increases slightly, probably because the

proportion of day with temperatures below 0 °C for the validation period is 0.21, which is lower than that for the calibration

period (0.24), but the precipitation during the days with temperature below 0 °C does not show obvious changes (2.62 mm·day$^{-1}$

for the calibration and 2.67 mm·day$^{-1}$ for the validation periods).




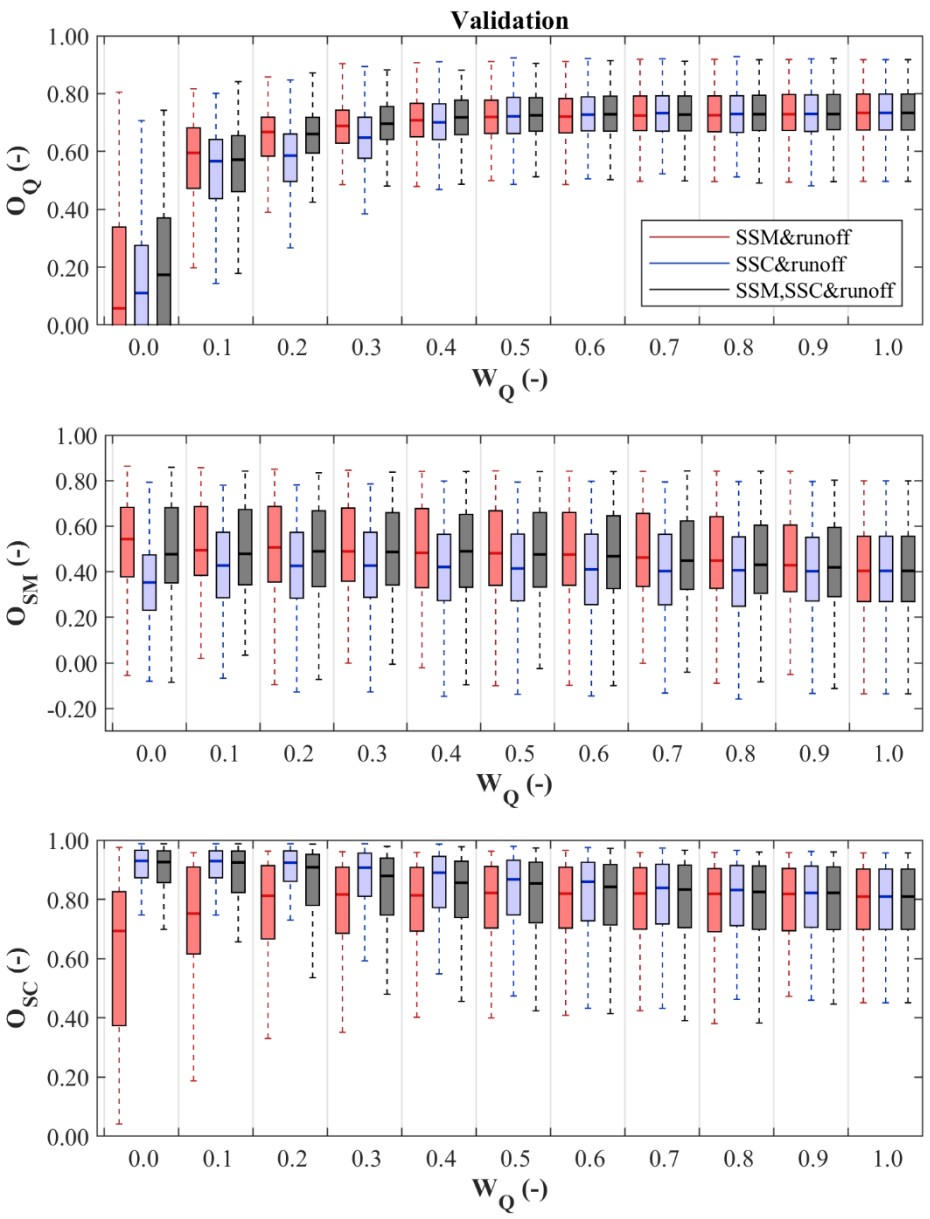

**Figure 4: Hydrologic model performance for three multiple objective calibration variants: calibration to satellite soil moisture and runoff (red boxes), to satellite snow cover and runoff (blue boxes) and to satellite soil moisture, snow cover and runoff (grey boxes). Top, middle and bottom panels show runoff (Eq. 2), soil moisture (Eq. 5) and snow cover (Eq. 6) model efficiency for different weights of the runoff objective in the validation period 2010-2014, respectively. $w_Q$ =1 represents calibration to runoff only. Size of the boxes represents the spatial variability across the 213 catchments.**





The correlation (in terms of the Pearson correlation coefficient) between model performance and selected catchment attributes
       (Table 2) is evaluated in Figures 5 and 6 in order to understand in which type of catchments SSM and SSC have the most
       relevant effect on model performance. The runoff model efficiency during the calibration period (Figure 5, left panel) increases
       with increasing mean number of days with negative air temperatures (MTL0, correlation over 0.57 for $w_Q$ larger than 0.4) and
       mean catchment elevation (MELE, correlation over 0.55 for $w_Q$ larger than 0.4) and tends to decrease with increasing
catchment mean annual air temperature (MAT, absolute correlation over 0.57 for $w_Q$ larger than 0.4). The larger runoff model
       efficiency in Alpine catchment than in the lowlands is likely related to the seasonality of snowmelt runoff which is easier to
       simulate than individual, more erratic events in the lowlands (Merz and Blöschl, 2009). The correlation of runoff model
       efficiency and catchment attributes increases with increasing runoff weight $w_Q$ and is not statistically significant or low (i.e.
       less 0.4) for $w_Q<0.4$ for most of the attributes. The correlations of the catchment attributes with soil moisture and snow
efficiencies are not consistently related to runoff weight. Soil moisture efficiency increases with increasing fraction of
       agricultural land (AP) where the correlation varies between 0.75 and 0.79 for different $w_Q$. This trend may be explained by the
       fact that soil moisture can generally be monitored more accurately in relatively flat, agricultural landscape than in rugged
       mountainous terrain (Brocca et al. 2013; Parajka et al., 2006), which in Austria are furthermore dominantly covered by forests
       and other dense vegetation impenetrable to the radar and scatterometer signals .Accordingly, we find the  soil moisture
efficiency tends to decrease with increasing forest cover (FP, correlation varies between -0.35 and -0.49). Also snow model
       efficiency tends to increase with decreasing MELE and SL (correlation is between -0.52 to -0.89), but increases with increasing
       MAT (correlations exceed 0.8 for most of the $w_Q$). In the flatlands, snow is less important so the cumulative number of days
       with potential snow errors in the objective function is generally lower.

       The correlations for the validation period (Figure 6) have the same pattern as for the calibration period (Fig.5). The attributes
with the largest correlations with runoff efficiency are the same and correlation tends to increase with increasing $w_Q$ as well.
       The correlation is generally only slightly lower than that estimated in the calibration period. The soil moisture efficiency in
       the validation period is positively correlated with AP (correlation = 0.76-0.79) and MAT (correlation = 0.55-0.68), but the
       correlation with AP is lower than in the calibration period. The largest negative correlation is found for calibration to runoff
       only ($w_Q=1$) and is larger than 0.7 for SL, FP, catchment elevation range (ER), and standard deviation of MAT and mean daily
potential global radiation (SDGR). The snow model efficiency is clearly related to topography, as it increases with decreasing
       MELE and increasing MAT.

       The relationship between model efficiencies and catchment attributes for the other two calibration variants are similar and are
       presented in Supplement, Figs S1-S4. The results show that including snow or soil moisture data in model calibration does not
       change the correlation between model efficiencies and catchment characteristics. The model efficiency is mainly related to
topography and certain climate, land cover and soil attributes, which are on the other hand cross-correlated with topography
       (Figure S1).







**Figure 5:** Correlation between catchment attributes (Table 2) and model performance, i.e. runoff (Eq. 2, left panel), soil moisture (Eq. 5, middle panel) and snow cover (Eq. 6, right panel), obtained from multiple objective calibration to satellite soil moisture (ASCAT), satellite snow cover (MODIS) and runoff (Var 3 of Table 5) in the calibration period 2000-2010. Cool and warm colors represent positive and negative correlations, respectively. Bold print indicates significance with p-value lower than 0.05.





**Figure 6:** Correlation between catchment attributes (Table 2) and model performance, i.e. runoff (Eq. 2, left panel), soil moisture (Eq. 5, middle panel) and snow cover (Eq. 6, right panel), obtained from multiple objective calibration to satellite soil moisture (ASCAT), satellite snow cover (MODIS) and runoff (Var 3 of Table 6) in the validation period 2010-2014. Cool and warm colors represent positive and negative correlations, respectively. Bold print indicates significance with p-value lower than 0.05.




## 4.2 Variability in calibrated model parameter values

Figure 7 compares the medians of the model parameters obtained with the three multiple calibration variants, grouped by snow, soil, runoff generation and runoff routing parameters in the columns from left to right. The snow-related parameters (left
column) are similar for the two calibration variants that use satellite snow cover, while the variant that uses soil moisture and runoff tends to have different values, particularly for the threshold temperature parameters (Tr, Ts, Tm). The medians of the snow correction (SCF) and melt (DDF) factors tend to be similar in all three variants if $w_Q > 0.4$.

The soil-related parameters (2$^{nd}$ column) show similar patterns. The variants that use satellite soil moisture in model calibration have more similar soil model parameter values than the one that uses only SSC and runoff. This suggests that adding soil
moisture satellite data in model calibration affects the soil-related parameters strongly and adding snow and soil moisture is complementary. The similarity of the variant using all three variables with those variants where alternatively SSC and SSM are left out suggests that SSC is more important for the snow-related parameters, and SSM is more important for the soil-related parameters, as would be expected. Increasing the runoff weight tends to decrease the difference between the calibration variants of the snow-related and the soil-related parameters. The runoff generation-related parameters (3$^{rd}$ column) tend to be
more similar for the two variants that use SSC, and the runoff routing-related parameters (right column) are always rather similar.

In a next step the model parameters obtained by multiple objective calibration are compared with those obtained by traditional calibration to runoff only (Fig. 8). The figure shows that the similarity between model parameters decreases with decreasing $w_Q$.

Snow-related parameters calibrated using SSC (middle column, top five lines) deviate quickly from those using runoff only as $w_Q$ decreases. Similarly, the soil-related parameters calibrated using SSM (left column, lines 6-8 from top) deviate quickly from their counterparts based only on runoff calibration. The difference in similarity/correlation between multiple objective calibration variants and runoff only calibration is smaller for runoff generation parameters. The runoff routing model parameters seem to be not very sensitive to selected model efficiencies and the correlation between model parameters is very
small.






**Figure 7: Medians of the parameter values from the three multiple objective calibration variants (lines) and different runoff weights $w_Q$ (Table 4). Red, blue and grey lines represent the calibration variants using soil moisture and runoff, snow cover and runoff, and soil moisture, snow cover data and runoff, respectively. Lines represent the median of the 213 Austrian catchments.**






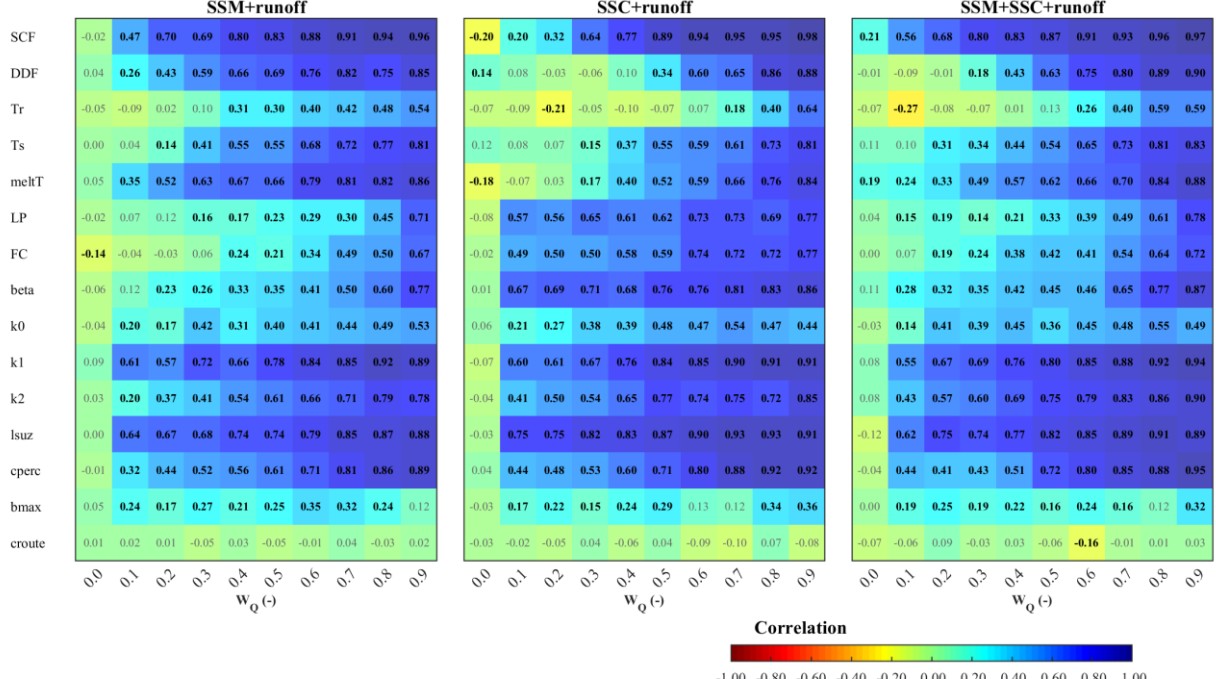

**Figure 8: Correlation of parameter values from three multiple objective calibration variants (runoff weights $w_Q$ = 0.0 to 0.9, Table 4) with those from traditional calibration to runoff only ($w_Q$=1.0). Left, middle and right panels represent calibration variants using soil moisture and runoff, snow cover and runoff and all three variables, respectively (Var 1, 2, 3 of Table 5 and 6). Cool and warm colors represent positive and negative correlations, respectively. Bold print indicates significance with p-value lower than 0.05.**

## 4.3 Comparison of multiple objective calibration to calibration to runoff only

The relative difference between the model efficiency of the three multi objective variants and that based on calibration to runoff only is presented in Figure 9. The runoff model efficiency of multiple objective calibration tends to be slightly lower than the traditional calibration to runoff only. The median of difference in runoff model efficiency of the two variants that uses SSM in model calibration is less than 3.2% for $w_Q$ larger than 0.3 in both calibration and validation periods. The multiple objective calibration to SSC and runoff has a somewhat larger median of difference for $w_Q$ between 0.2 and 0.4, but for larger $w_Q$ the median is almost identical with that of the other multiple objective variants. The integration of soil moisture in model calibration improves the correlation of satellite and simulated soil moisture. The median improvement for $w_Q$ <0.6 is larger than 30% and 15% in the calibration and validation periods, respectively. The calibration to all variables has a median relative improvement about 4% to 25% lower than the calibration to SSM and runoff in the calibration period, but is very similar in the validation period. The calibration to SSM and runoff does not improve snow cover simulations, but the use of all variables





improves the snow model efficiency. For $w_Q$ less than 0.5 the median improvement is larger than 5% in both calibration and validation periods.

Given that the median runoff efficiency is not improved by the addition of soil moisture and snow data (Figure 9) it is of interest to see how the changes are distributed in space. Figure 10 shows that in up to 40% of catchments the validation runoff efficiency is improved by using multiple objective calibration as compared to calibration to runoff alone. The number of catchments with runoff improvements increases with increasing runoff weight. The flipside of course is that in the remaining catchments the runoff model efficiency is deteriorated. The calibration variants that use SSM data improve soil moisture

simulations in more than 80% of the catchments for all weights, and the addition of snow data does not change the performance. The snow model efficiency is vastly improved by the inclusion of the SSC data, for $w_Q$>0.6 in almost all catchments, and the inclusion of soil moisture (in the variant that uses all variables) has still a very big improvement of snow simulations as compared to the case when only runoff is used in the calibration.

Overall, there are two important messages. The inclusion of soil moisture data in the calibration mainly improves the soil

moisture simulations; the inclusion of snow data in the calibration mainly improves the snow simulations; and including both of them improves both soil moisture and snow simulations to a similar extent. Second, when comparing the panels of Figure 10, one sees that the snow data are more efficient in improving snow simulations than the soil moisture data are in improving soil moisture simulations.




**Figure 9: Relative difference in model efficiency of three multiple objective calibration variants (lines) using different runoff weights**
$w_Q$ **(Table 4) compared to traditional calibration to runoff only (**$w_Q$**=1). Red, blue and grey lines represent calibration variants using soil moisture and runoff, snow cover and runoff, and soil moisture, snow cover and runoff, respectively Lines represent the median of the 213 Austrian catchments. Top, middle and bottom panels refer to runoff, soil moisture and snow cover efficiencies in the**

**calibration (left panels) and validation (right panels) periods, respectively.**





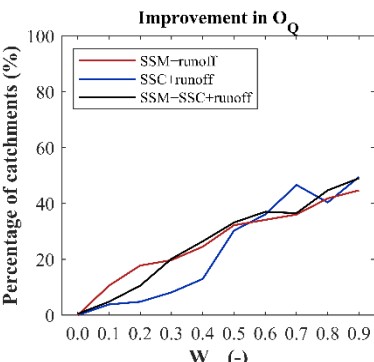 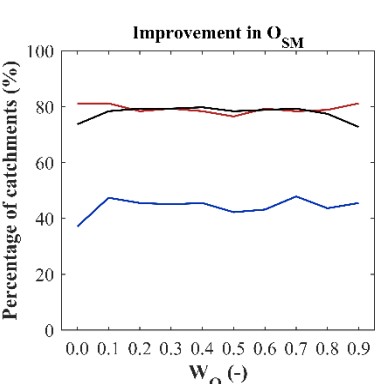 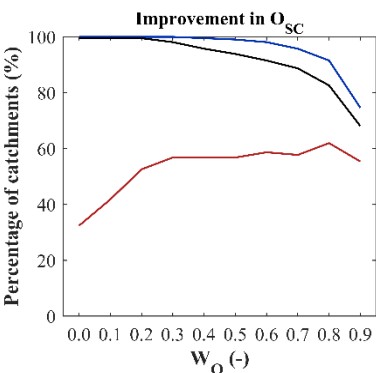

**Figure 10. Relative number of catchments with improvement in runoff (left), soil moisture (centre) and snow cover (right) model efficiency in the validation period. Relative number relates to the 213 Austrian catchments used in this study.**

It is now of interest to see whether the catchments for which the runoff and soil moisture model efficiencies are improved by
the inclusion of SSC and SSM data (Figure 10) are different from those for which this is not the case. The distribution of the catchment attributes of these two catchment groups are therefore compared with the Kolmogorov-Smirnov two sample test (KS). Since the snow model efficiency is improved in almost all study catchments, it is not analysed here. Table 7 and Table 8 show the p–values of the KS for the runoff and soil moisture model efficiencies, respectively, indicating statistically significant differences between the catchment groups in many cases. For catchments with a large frequency of runoff
improvements (e.g. for $w_Q = 0.7$ and 0.8) there are a number of differentiating factors including those related to topography (mean catchment elevation, mean catchment slope), proportion of agricultural land, mean annual air temperature, number of days with negative air temperature and mean saturated hydraulic conductivity. An example of differences between the groups in terms of mean catchment elevation (MELE) and percentage of arable land (AP) is presented in Fig. 11. The results indicate that improvement in runoff is observed in catchments with lower mean catchment elevation and a larger proportion of
agricultural land. The other catchment attributes with statistically significant differences are correlated with MELE, so have similar differences in the distributions as those presented in Fig. 11.

Since an improvement in soil moisture simulations is observed in 80% of the catchments (Fig. 10) their attributes are particularly interesting. The factors controlling the improvement include topographical (MELE, SL, ER, SDGR), land cover (FP, AP), climate (MAP, SDAP, MAT, CAI, MTLT0) and soil (MKS) attributes, similarly as for improvement in runoff. This
is illustrated in Fig. 12 that indicates that improvement in soil moisture simulations occurs particularly in catchments with low mean catchment elevation and a large proportion of agricultural land. In contrast to the runoff improvements, the results for the improvement in soil moisture are not related to the runoff weight $w_Q$ used in the model calibration.



**Figure 11: Distributions of mean catchment elevation (MELE) and percentage of arable land (AP) for the groups of catchments with (blue) and without (red) runoff model efficiency improvement in the validation period when including soil moisture and snow data in the calibration.**





**Figure 12: Distributions of mean catchment elevation (MELE) and percentage of arable land (AP) for the groups of catchments with**
**(blue) and without (red) soil moisture model efficiency improvement in the validation period when including soil moisture and snow**
**data in the calibration**



# 5 Discussion and conclusions

In this study, we tested three multiple calibration variants using runoff data along with ASCAT SWI soil moisture and MODIS snow cover data. The calibration runoff model efficiency is similar to previous studies that used only runoff for model calibration. For example the median of runoff model efficiency ranges between 0.77-0.79 (for runoff weights larger than 0.4) which is similar to the medians of 0.80 and 0.84 found in Parajka et al. (2008, 2009) for 148 Austrian catchments, and better than the median of 0.76 found for 320 Austrian catchments in Parajka et al., (2006) as well as the range 0.70-0.73 found for

the same set of catchments as in this paper but using a lumped model (Sleziak et al., 2018).

Results show that the inclusion of satellite soil moisture data in the calibration mainly improves the soil moisture simulations. The median soil moisture correlation between hydrologic model outputs and ASCAT SWI is 0.4 to 0.52 (depending on the weight $w_Q$), which is significantly larger than the median of 0.26 found by using the coarser ERS scatterometer data in model calibration in Parajka et al. (2006). This reflects improvements both in the instrument specifications (better temporal and spatial

sampling, higher radiometric accuracy, etc.) and the retrieval algorithm (Naeimi et al. 2009, Hahn et al. 2020).

The inclusion of satellite snow data mainly improves the snow simulations. Using MODIS snow cover data to constrain the model parameters shows strong ability in improving the accuracy of representing the snow accumulating and melting processes from the model. When giving weights $w_Q<0.5$ to snow, almost all the catchments showed improvements in snow cover simulations. In terms of the improvement in snow model efficiency, our results are better than the results from Parajka et al.

(2008), in their study only 3 years MODIS snow cover data was used, the improvement of snow mapping even depended on the data availability.

The satellite snow data are more efficient in improving snow simulations than the satellite soil moisture data are in improving soil moisture simulations. Part of the reason may be related to problems in mapping soil moisture in the alpine region while MODIS snow cover is very accurate both in the lowlands and in the mountains. For example, Parajka and Blöschl (2006) and

Tong et al. (2019) showed the classification accuracy of the MODIS snow cover range from 95% to over 97% in Austria. Furthermore, it is interesting that including both soil moisture and snow cover data improves both soil moisture and snow simulations to almost the same extent as if including them individually, without any significant deterioration in the other variable. This gives the possibility to consistently improve the simulations of snow and soil moisture in future model applications. Our validation results indicate that snow simulations are improved in almost all, soil moisture correlation in about

80% and runoff in up to 40% of the catchments. Overall, the runoff performance changes very little when including soil moisture and snow data in the calibration.

The calibrated snow-related parameters are strongly affected by including snow data, and to a lesser extent by soil moisture data, while the soil-related parameters are only affected by soil moisture data. This separation is a welcome property as it facilitates parameter calibration. The soil moisture data also have some effect on the snow-related parameters. As the melting

changes the soil moisture directly, the soil moisture data provide additional constraints on the parameters controlling snowmelt. This can be helpful in understanding hydrological processes, especially for the variation of snow water equivalent.



Our results indicate that the runoff and soil moisture simulation improvement when including soil moisture data in the calibration is found mainly in catchments with lower mean catchment elevation and a larger proportion of agricultural land. While, overall, in 40% of the catchments the validation runoff efficiency is improved by the inclusion of soil moisture (Figure
10, left panel), these are about 50% of the catchments if only those with elevation lower than the median (1011 m a.s.l) and agricultural area larger than its median (16.3%) are considered. Similarly, while, overall, in 80% of the catchments the validation soil moisture efficiency is improved by the inclusion of soil moisture (Figure 10, middle panel), these are about 90% of the catchments if only those which low elevation and agricultural use are considered. The higher efficiency in improving the hydrologic model in the lowlands can be explained by the better quality of the ASCAT soil moisture retrievals
(compared to the alpine regions), but is likely also due to the higher spatial consistency in soil texture and land cover type, and also lower slope and elevation variation. In contrast to a previous assessment of ERS assimilation into model calibration (Parajka et al., 2006), we found soil moisture improvement not only in lowland catchments with lower topographical variability, but also in catchments with smaller sizes (Figure 5) which may be related to the higher spatial and temporal resolution of ASCAT as compared to ERS. Over flatlands, ASCAT retrievals have improved a lot compared to the ERS
retrievals 15 years back, but in alpine regions, the rugged topography, dense alpine vegetation, and presence of snow and ice even during the summer, makes using the data still challenging, given the higher retrieval errors and invalid measurements when the ground is snow covered or frozen. Additionally, the large heterogeneity in temperature and snow cover in mountainous regions can lead to insufficient masking for frozen soil and snow cover.

This study has evaluated the potential of assimilating the soil water index (representing root zone soil moisture) into hydrologic
model calibration. It would be useful to extend this study to assimilate other variables, such as surface soil moisture estimates by using a dual soil moisture conceptual hydrologic model (Parajka et al., 2009) and also compare the role of the spatial resolution of soil moisture and snow data on their value in the assimilation.

**Data availability**

The discharge data from the HZB can be accessed through https://ehyd.gv.at/ (BMLRT, 2020). The meteorological data from the ZAMG are currently not freely available, requests should be directed to klima@zamg.ac.at. The ASCAT soil moisture data is available via Copernicus Global Land Service (https://land.copernicus.eu/). MODIS C6 snow cover products are from NASA National Snow & Ice Data Center (https://nsidc.org/). Processed ASCAT SWI and MODIS snow cover maps used in this study are available upon request. Landuse information is from Copernicus Land Monitoring Service (https://land.copernicus.eu/).
MODIS C6 Normalized Difference Vegetation Index (MOD13A3v006) is from NASA EOSDIS Land Processes DAAC (https://doi.org/10.5067/MODIS/MOD13A3.006). The maps of Soil hydraulic properties are from Zhang et al., (2018, https://doi.org/10.7910/DVN/UI5LCE). The R package of TUWmoel can be downloaded from CRAN (https://cran.r-project.org/web/packages/TUWmodel/index.html).

**Author contribution**



RT and JP conceived and designed the study, wrote the codes, performed the analyses, and prepared the manuscript. AS and IP prepared the ASCAT soil moisture SWI data used in this study. JK, BS, MK and PV were responsible for the data management, including quality control, processing, and validating. MV and WW provided the analyses related to soil moisture and prepared part of the manuscript. GB supervised the study and contributed to the study design and interpretation of the results. All authors took part in the discussion of the results and revisions of the paper.

**Competing interests**

The authors declare that they have no conflict of interest.

**Acknowledgment**

The authors would like to acknowledge financial support provided by the Austrian Science Funds (FWF) as part of the Vienna Doctoral Program on Water Resource Systems (DK W1219-N28) and the Austrian Research Promotion Agency (FFG) through the BMon project (Contract No. 866031). Rui Tong is grateful for the scholarship from China Scholarship Council (CSC).

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



**Table 1. Normalized Difference Snow Index (NDSI) threshold for snow cover mapping from Tong et al. (2019)**

| Aqua | | Jan | Feb | Mar | Apr | May | Jun | Jul | Aug | Sep | Oct | Nov | Dec |
|---|---|---|---|---|---|---|---|---|---|---|---|---|---|
| below | non-forest | 0.36 | 0.34 | 0.36 | 0.67 | 0.98 | 0.97 | 0.89 | 0.96 | 0.84 | 0.67 | 0.43 | 0.41 |
| 900 m | coniferous forest | 0.17 | 0.24 | 0.34 | 0.50 | 0.63 | 0.66 | 0.79 | 0.77 | 0.92 | 0.47 | 0.42 | 0.26 |
| a.s.l. | other forest | 0.32 | 0.29 | 0.29 | 0.82 | 0.70 | 0.78 | 0.75 | 0.89 | 0.90 | 0.80 | 0.47 | 0.30 |
| over | non-forest | 0.22 | 0.18 | 0.28 | 0.49 | 0.67 | 0.91 | 0.89 | 0.91 | 0.71 | 0.43 | 0.45 | 0.26 |
| 900 m | coniferous forest | 0.20 | 0.14 | 0.29 | 0.49 | 0.74 | 0.86 | 0.82 | 0.87 | 0.82 | 0.49 | 0.43 | 0.25 |
| a.s.l. | other forest | 0.17 | 0.12 | 0.13 | 0.50 | 0.83 | 0.74 | 0.77 | 0.58 | 0.69 | 0.44 | 0.46 | 0.20 |

| Terra | | Jan | Feb | Mar | Apr | May | Jun | Jul | Aug | Sep | Oct | Nov | Dec |
|---|---|---|---|---|---|---|---|---|---|---|---|---|---|
| below | non-forest | 0.32 | 0.30 | 0.37 | 0.57 | 0.74 | 0.78 | 0.82 | 0.85 | 0.82 | 0.52 | 0.41 | 0.36 |
| 900 m | coniferous forest | 0.30 | 0.19 | 0.29 | 0.34 | 0.72 | 0.80 | 0.65 | 0.93 | 0.61 | 0.56 | 0.31 | 0.34 |
| a.s.l. | other forest | 0.22 | 0.19 | 0.29 | 0.86 | 0.78 | 0.64 | 0.78 | 0.74 | 0.65 | 0.65 | 0.40 | 0.31 |
| over | non-forest | 0.21 | 0.16 | 0.20 | 0.44 | 0.60 | 0.90 | 0.77 | 0.70 | 0.84 | 0.54 | 0.35 | 0.27 |
| 900 m | coniferous forest | 0.16 | 0.12 | 0.22 | 0.40 | 0.65 | 0.78 | 0.80 | 0.76 | 0.78 | 0.47 | 0.37 | 0.20 |
| a.s.l. | other forest | 0.10 | 0.11 | 0.10 | 0.17 | 0.53 | 0.16 | 0.26 | 0.69 | 0.67 | 0.31 | 0.31 | 0.22 |






**Table 2. Statistics of the catchment attributes of the 213 catchments in Figure 1 with abbreviation, unit, minimum, maximum, and median. The standard deviations refer to spatial variability within each catchment.**

| Information | Attribute | Abbrev. | Unit | Min. | Max. | Median |
|---|---|---|---|---|---|---|
| Size | Area | A | km$^2$ | 13.70 | 6214.00 | 167.30 |
| Elevation | Mean elevation | MELE | m a.s.l. | 353.01 | 2939.76 | 1010.73 |
| | Minimum elevation | MiELE | m a.s.l. | 200.00 | 1939.00 | 561.00 |
| | Maximum elevation | MxELE | m a.s.l. | 509.00 | 3760.00 | 1861.00 |
| | Elevation range | ER | m | 80.00 | 3072.00 | 1279.00 |
| | Roughness index (MELE-MiELE)/ER | RI | - | 0.15 | 0.65 | 0.38 |
| | Mean slope | SL | % | 1.74 | 43.91 | 18.84 |
| | Mean daily potential global radiation | MGR | kW·m$^{-2}$·day | 4.73 | 6.26 | 5.19 |
| | Standard deviation of MGR | SDGR | kW·m$^{-2}$·day | 0.02 | 1.10 | 0.39 |
| Land cover | Coverage of forest | FP | % | 0.00 | 94.59 | 46.88 |
| | Coverage of agricultural areas | AP | % | 0.00 | 92.86 | 16.30 |
| | Mean monthly normalized difference vegetation index | MNDVI | - | 0.00 | 0.71 | 0.60 |
| | Standard deviation of MNDVI | SDNDVI | - | 0.02 | 0.19 | 0.06 |
| Climate | Mean annual precipitation | MAP | mm | 728.13 | 2301.84 | 1274.40 |
| | Standard deviation of annual MAP | SDAP | mm | 10.79 | 367.57 | 124.70 |
| | Mean air temperature | MAT | °C | -2.83 | 10.30 | 7.36 |
| | Standard deviation of MAT | SDAT | °C | 0.06 | 3.55 | 1.26 |
| | Mean annual potential evaporation | MEPI | mm | 233.49 | 740.45 | 629.57 |
| | Standard deviation of MEPI | SDEPI | mm | 4.33 | 162.07 | 60.17 |
| | Catchment aridity index (MEPI/MAP) | CAI | - | 0.18 | 0.98 | 0.47 |
| | Standard deviation of aridity index | SDAI | - | 0.01 | 0.31 | 0.08 |
| | Proportion of day with temperature below 0 °C | MTL0 | - | 0.12 | 0.62 | 0.20 |
| Soil | Mean field capacity | MFC | cm$^3$·cm$^{-3}$ | 0.29 | 0.43 | 0.36 |
| | Standard deviation of MFC | SDFC | cm$^3$·cm$^{-3}$ | 0.01 | 0.05 | 0.02 |
| | Mean saturated hydraulic conductivity | MKS | cm·day$^{-1}$ | 24.88 | 327.77 | 161.17 |
| | Standard deviation of MKS | SDKS | cm·day$^{-1}$ | 6.43 | 76.03 | 40.35 |






**Table 3. Parameters of the hydrologic model (TUWmodel) and ranges used in model calibration. A suitable parameter range may vary by climate and land cover regions and is usually set by expert judgement. The range used here is based on the previous experience of Merz et al. (2011) and Viglione et al. (2013) in the study area.**

| Parameter | Explanation [Unit] | General Range |
|---|---|---|
| SCF | snow correction factor [-] | 0.9-1.5 |
| DDF | degree day factor [mm/ °C/day] | 0.0-6.0 |
| Ts | threshold temperature below which precipitation is snow [°C] | -3.0-1.0 |
| Tr | threshold temperature above which precipitation is rain [°C] | 1.0-3.0 |
| meltT | threshold temperature above which melt starts [°C] | -2.0-2.0 |
| LP | parameter related to the limit for potential evaporation [-] | 0.0-1.0 |
| FC | field capacity, i.e., max soil moisture storage [mm] | 0-600 |
| beta | the nonlinear parameter for runoff production [-] | 0-20 |
| k0 | storage coefficient for very fast response [day] | 0-2.0 |
| k1 | storage coefficient for fast response [day] | 2-30 |
| k2 | storage coefficient for slow response [day] | 30-250 |
| lsuz | threshold storage state, i.e., the very fast response starts if exceeded [mm] | 1-100 |
| cperc | constant percolation rate [mm/day] | 0.0-8.0 |
| bmax | maximum base at low flows [days] | 0.0-30.0 |
| croute | free scaling parameter [days$^2$/mm] | 0.0-50.0 |






**Table 4. Weights given to runoff, satellite soil moisture (SSM) and satellite snow cover (SSC) in the multiple objective calibration (Eq.1) for different calibration variants. a set of 11 $w_Q$ weights in the range 0.0 and 1.0 is tested for each multiple objective calibration variant. The sum of weights is always 1.0.**

| Calibration variant | Weight of runoff ($w_Q$) | Weight of soil moisture ($w_{SM}$) | Weight of snow cover ($w_{SC}$) |
|---|---|---|---|
| Runoff only | $w_Q = 1.0$ | $w_{SM} = 0.0$ | $w_{SC} = 0.0$ |
| SSM+runoff (Var1) | $w_Q = \{k/10\}_{k=0}^{10}$ | $w_{SM} = 1 - w_Q$ | $w_{SC} = 0.0$ |
| SSC+runoff (Var2) | $w_Q = \{k/10\}_{k=0}^{10}$ | $w_{SM} = 0.0$ | $w_{SC} = 1 - w_Q$ |
| SSM+SSC+runoff (Var3) | $w_Q = \{k/10\}_{k=0}^{10}$ | $w_{SM} = w_{SC}$ | $w_{SC} = w_{SM}$ |






**Table 5. Median of runoff (Eq. 2), soil moisture (Eq. 5) and snow cover (Eq. 6) model efficiency obtained from three multiple objective calibration variants: (1) to satellite soil moisture (ASCAT) and runoff (var1); (2) to satellite snow cover (MODIS) and runoff (var2); (3) to satellite soil moisture (ASCAT), satellite snow cover (MODIS) and runoff (var3) in 213 catchments in the calibration period 2000-2010.**

| Weight $w_Q$ | Runoff model efficiency | | | Soil moisture efficiency | | | Snow cover efficiency | | |
|---|---|---|---|---|---|---|---|---|---|
| | Var1 | Var2 | Var3 | Var1 | Var2 | Var3 | Var1 | Var2 | Var3 |
| 0.00 | 0.07 | 0.19 | 0.18 | 0.52 | 0.26 | 0.45 | 0.66 | 0.91 | 0.91 |
| 0.10 | 0.65 | 0.60 | 0.59 | 0.49 | 0.30 | 0.44 | 0.75 | 0.91 | 0.91 |
| 0.20 | 0.71 | 0.62 | 0.69 | 0.47 | 0.32 | 0.43 | 0.80 | 0.91 | 0.88 |
| 0.30 | 0.74 | 0.70 | 0.73 | 0.46 | 0.32 | 0.43 | 0.80 | 0.88 | 0.86 |
| 0.40 | 0.75 | 0.74 | 0.76 | 0.44 | 0.31 | 0.42 | 0.80 | 0.87 | 0.84 |
| 0.50 | 0.76 | 0.77 | 0.77 | 0.43 | 0.31 | 0.42 | 0.80 | 0.85 | 0.83 |
| 0.60 | 0.77 | 0.78 | 0.78 | 0.43 | 0.30 | 0.40 | 0.80 | 0.84 | 0.82 |
| 0.70 | 0.78 | 0.78 | 0.78 | 0.41 | 0.30 | 0.37 | 0.80 | 0.82 | 0.81 |
| 0.80 | 0.78 | 0.79 | 0.79 | 0.39 | 0.29 | 0.36 | 0.80 | 0.81 | 0.80 |
| 0.90 | 0.79 | 0.79 | 0.79 | 0.34 | 0.30 | 0.32 | 0.80 | 0.80 | 0.80 |
| 1.00 | 0.79 | 0.79 | 0.79 | 0.29 | 0.29 | 0.29 | 0.79 | 0.79 | 0.79 |




**Table 6. Median of runoff (Eq. 2), soil moisture (Eq. 5) and snow cover (Eq. 6) model efficiency obtained from three multiple objective calibration variants: (1) to satellite soil moisture (ASCAT) and runoff (var1); (2) to satellite snow cover (MODIS) and runoff (var2); (3) to satellite soil moisture (ASCAT), satellite snow cover (MODIS) and runoff (var3) in 213 catchments in the validation period 2010-2014.**

| Weight | Runoff model efficiency | | | Soil moisture efficiency | | | Snow cover efficiency | | |
|---|---|---|---|---|---|---|---|---|---|
| | Var1 | Var2 | Var3 | Var1 | Var2 | Var3 | Var1 | Var2 | Var3 |
| 0.00 | 0.06 | 0.11 | 0.17 | 0.54 | 0.35 | 0.48 | 0.69 | 0.93 | 0.93 |
| 0.10 | 0.60 | 0.57 | 0.57 | 0.49 | 0.43 | 0.48 | 0.75 | 0.93 | 0.92 |
| 0.20 | 0.67 | 0.59 | 0.66 | 0.51 | 0.43 | 0.49 | 0.81 | 0.92 | 0.91 |
| 0.30 | 0.69 | 0.65 | 0.70 | 0.49 | 0.43 | 0.49 | 0.82 | 0.91 | 0.88 |
| 0.40 | 0.71 | 0.70 | 0.72 | 0.48 | 0.42 | 0.49 | 0.81 | 0.89 | 0.86 |
| 0.50 | 0.72 | 0.72 | 0.72 | 0.48 | 0.41 | 0.48 | 0.82 | 0.87 | 0.85 |
| 0.60 | 0.72 | 0.73 | 0.73 | 0.48 | 0.41 | 0.47 | 0.82 | 0.86 | 0.84 |
| 0.70 | 0.72 | 0.73 | 0.73 | 0.46 | 0.40 | 0.45 | 0.82 | 0.84 | 0.83 |
| 0.80 | 0.73 | 0.73 | 0.73 | 0.45 | 0.41 | 0.43 | 0.82 | 0.83 | 0.83 |
| 0.90 | 0.73 | 0.73 | 0.73 | 0.43 | 0.40 | 0.42 | 0.82 | 0.82 | 0.82 |
| 1.00 | 0.73 | 0.73 | 0.73 | 0.40 | 0.40 | 0.40 | 0.81 | 0.81 | 0.81 |



**Table 7. Kolmogorov-Smirnov p-values testing the similarity of the distribution of catchment attributes across the 213 catchments between those catchments where the runoff model efficiency is improved in the validation period by the inclusion of the soil moisture and snow data in the calibration and those catchments where this is not the case. The null hypothesis that the two samples were drawn from the same distribution is rejected if the p-value is less than the significance level (bold print).**

| $w_Q$ | 0 | 0.1 | 0.2 | 0.3 | 0.4 | 0.5 | 0.6 | 0.7 | 0.8 | 0.9 |
|---|---|---|---|---|---|---|---|---|---|---|
| A | 0.70 | 0.34 | 0.22 | 0.32 | 0.15 | 0.93 | **0.05** | 0.26 | 0.46 | 0.96 |
| MELE | 0.23 | **0.00** | **0.00** | **0.01** | **0.01** | **0.00** | 0.17 | **0.03** | 0.05 | 0.25 |
| MiELE | 0.27 | 0.06 | 0.09 | 0.20 | **0.03** | **0.01** | **0.03** | **0.01** | 0.09 | 0.19 |
| MxELE | 0.60 | **0.00** | **0.00** | **0.00** | **0.01** | **0.00** | 0.23 | **0.03** | **0.03** | 0.35 |
| ER | 0.59 | **0.00** | **0.00** | **0.00** | **0.00** | **0.00** | 0.47 | 0.29 | **0.02** | 0.21 |
| RI | 0.49 | **0.00** | **0.00** | **0.00** | **0.01** | **0.00** | 0.14 | **0.03** | **0.03** | 0.05 |
| SL | 0.11 | 0.46 | 0.82 | 0.37 | 0.50 | 0.16 | 0.18 | 0.17 | 0.86 | 0.81 |
| MGR | 0.23 | 0.73 | 0.38 | 0.62 | 0.05 | 0.66 | 0.48 | 0.98 | 1.00 | 0.45 |
| SDGR | 0.44 | **0.00** | **0.01** | **0.00** | **0.00** | **0.00** | 0.11 | 0.07 | **0.02** | 0.06 |
| FP | 0.29 | 0.05 | **0.05** | 0.07 | 0.40 | 0.23 | 0.87 | 0.80 | 0.31 | 0.72 |
| AP | 0.20 | **0.00** | **0.00** | **0.00** | **0.00** | **0.00** | 0.08 | **0.02** | **0.02** | 0.17 |
| MNDVI | 0.71 | **0.01** | 0.10 | **0.03** | **0.04** | **0.04** | **0.05** | **0.04** | 0.15 | 0.21 |
| SDNDVI | 0.72 | **0.00** | **0.04** | **0.00** | **0.00** | **0.01** | 0.49 | 0.14 | **0.01** | 0.09 |
| MAP | 0.82 | **0.00** | 0.19 | 0.08 | 0.51 | **0.01** | **0.03** | 0.12 | **0.01** | **0.05** |
| SDAP | 0.34 | **0.00** | 0.05 | **0.04** | 0.15 | 0.38 | 0.93 | 0.92 | **0.03** | 0.39 |
| MAT | 0.20 | **0.00** | **0.00** | **0.00** | **0.01** | **0.00** | 0.11 | **0.03** | 0.10 | 0.42 |
| SDAT | 0.43 | **0.00** | **0.01** | **0.01** | **0.00** | **0.01** | 0.55 | 0.16 | **0.02** | 0.21 |
| MEPI | 0.15 | **0.00** | **0.00** | **0.00** | **0.00** | **0.00** | 0.07 | **0.00** | 0.13 | 0.26 |
| SDEPI | 0.36 | **0.00** | **0.02** | **0.00** | **0.00** | **0.01** | 0.63 | 0.18 | **0.01** | 0.06 |
| CAI | 0.56 | **0.00** | **0.01** | **0.00** | 0.05 | **0.00** | 0.20 | **0.05** | **0.00** | **0.03** |
| SDAI | 0.78 | 0.06 | **0.03** | **0.04** | **0.01** | 0.29 | 0.38 | 0.74 | 0.91 | 0.82 |
| MTL0 | 0.20 | **0.00** | **0.00** | **0.00** | **0.01** | **0.00** | 0.19 | **0.02** | 0.11 | 0.31 |
| MFC | 0.67 | 0.14 | **0.03** | 0.30 | 0.38 | 0.08 | 0.10 | 0.16 | **0.03** | 0.10 |
| SDFC | 0.14 | 0.47 | 0.24 | 0.56 | **0.04** | 0.25 | 0.12 | 0.69 | **0.01** | **0.01** |
| MKS | 0.18 | **0.00** | **0.00** | **0.00** | **0.00** | **0.00** | 0.17 | **0.01** | 0.06 | 0.14 |
| SDKS | 0.40 | 0.19 | 0.38 | 0.98 | 0.07 | **0.05** | 0.47 | 0.55 | 0.11 | 0.27 |



**Table 8. Same as Table 7, but for the soil moisture model efficiency.**

| $w_Q$ | 0 | 0.1 | 0.2 | 0.3 | 0.4 | 0.5 | 0.6 | 0.7 | 0.8 | 0.9 |
|---|---|---|---|---|---|---|---|---|---|---|
| A | 0.53 | 0.31 | 0.18 | 0.61 | 0.23 | 0.26 | 0.45 | 0.98 | 0.98 | 0.26 |
| MELE | **0.01** | **0.01** | **0.02** | 0.07 | 0.09 | **0.05** | 0.05 | 0.70 | 0.14 | 0.94 |
| MiELE | **0.01** | 0.15 | 0.13 | 0.13 | 0.20 | **0.04** | 0.05 | 0.54 | 0.20 | 0.74 |
| MxELE | **0.01** | **0.00** | **0.01** | **0.01** | **0.01** | **0.00** | **0.01** | 0.13 | **0.02** | 0.31 |
| ER | 0.07 | 0.08 | 0.08 | **0.03** | **0.04** | **0.02** | 0.06 | 0.22 | 0.10 | 0.35 |
| RI | **0.01** | **0.03** | **0.01** | **0.03** | **0.03** | **0.01** | **0.02** | 0.38 | 0.05 | 0.34 |
| SL | 0.32 | **0.02** | 0.50 | 0.93 | 0.35 | 0.73 | 0.90 | 0.45 | 0.80 | 0.46 |
| MGR | 0.59 | 0.55 | 0.69 | 0.43 | 0.79 | 0.25 | 0.50 | 0.61 | 0.49 | 0.81 |
| SDGR | **0.00** | **0.01** | **0.01** | **0.01** | **0.05** | **0.02** | **0.03** | 0.14 | 0.07 | 0.17 |
| FP | **0.00** | **0.00** | **0.00** | **0.00** | **0.01** | **0.02** | 0.10 | 0.58 | **0.05** | 0.35 |
| AP | **0.00** | **0.04** | **0.01** | **0.04** | **0.04** | **0.00** | **0.00** | 0.41 | 0.12 | 0.32 |
| MNDVI | 0.23 | **0.01** | 0.25 | 0.89 | 0.97 | 0.58 | 0.36 | 0.89 | 0.62 | 0.67 |
| SDNDVI | 0.06 | 0.63 | 0.87 | 0.78 | 0.62 | 0.27 | 0.11 | 0.69 | 0.56 | 0.59 |
| MAP | **0.00** | **0.02** | **0.01** | **0.00** | 0.17 | **0.01** | **0.03** | 0.23 | 0.07 | 0.72 |
| SDAP | **0.03** | 0.14 | 0.07 | **0.00** | **0.01** | **0.01** | **0.00** | **0.01** | **0.02** | 0.07 |
| MAT | **0.01** | **0.02** | **0.01** | **0.03** | **0.05** | **0.02** | **0.02** | 0.40 | 0.12 | 1.00 |
| SDAT | **0.04** | 0.05 | 0.07 | 0.08 | 0.09 | 0.05 | 0.07 | 0.19 | 0.36 | 0.22 |
| MEPI | **0.00** | **0.00** | **0.04** | **0.04** | 0.06 | **0.02** | **0.02** | 0.32 | 0.18 | 1.00 |
| SDEPI | **0.03** | 0.06 | **0.03** | **0.04** | **0.05** | **0.03** | **0.05** | 0.34 | 0.17 | 0.32 |
| CAI | **0.01** | 0.05 | **0.01** | **0.01** | **0.04** | **0.01** | **0.02** | 0.39 | 0.07 | 0.82 |
| SDAI | 0.44 | 0.95 | 0.59 | 0.79 | 0.52 | 0.40 | 0.44 | 0.05 | 0.75 | **0.04** |
| MTL0 | **0.01** | **0.02** | **0.02** | 0.06 | 0.07 | **0.02** | **0.02** | 0.49 | 0.26 | 0.83 |
| MFC | **0.00** | **0.00** | **0.00** | **0.01** | **0.04** | **0.01** | **0.04** | 0.36 | 0.09 | 0.47 |
| SDFC | 0.19 | 0.47 | 0.34 | 0.15 | 0.24 | 0.05 | 0.86 | 0.97 | 0.16 | 0.76 |
| MKS | **0.03** | **0.00** | **0.05** | **0.05** | 0.06 | **0.02** | **0.04** | 0.53 | 0.18 | 0.76 |
| SDKS | **0.04** | 0.19 | 0.16 | **0.04** | 0.30 | 0.08 | 0.22 | 0.26 | 0.42 | 0.58 |
