# Peer review of "The value of ASCAT soil moisture and MODIS snow cover data for calibrating a conceptual hydrologic model"

_Hydrology and Earth System Sciences, 2020_

## Referee Comment (RC1) · Anonymous Referee #1 · 23 Oct 2020

Comments on "The value of ASCAT soil moisture and MODIS snow cover data for calibrating a conceptual hydrologic model"

The authors calibrated a conceptual hydrological model by using ASCAT soil moisture and MODIS snow cover data jointly or separately, and improvements of related variants have been achieved in varying degrees. The efficiency improvement was also analyzed under different scenarios and catchment attributes. Overall, the results seem convincing and the study is valuable for related research. However, there are several issues that still exist and need to be clarified further as indicated in the following.

First, the manuscript needs further editorial work to improve the paragraph structure

and some vague expressions. A single sentence definitely cannot be a paragraph (e.g., line 286), and a paragraph should not be too long or too short. In addition, please pay attention to vague expressions in this manuscript, such as line 84 " to compare the multiple objective calibration to soil moisture and runoff to three different calibration variants", which is really confusing. There are other similar sentences, so I hope the authors make a thorough change to improve the clarity of the manuscript.

Another major issue in this manuscript is that conclusions in the Results section can be presented in a more straightforward way. At its current form, the conclusions are over detailed and have too many numbers, which do not have much value. It is hard for the readers to get the key messages from the authors. Furthermore, the figures and tables contain too much information (e.g., Fig5-6, Table 7-8), also leading to the difficulty in deriving the key information. So please make more concise and clearer conclusions, and improve the presentation of all figures and tables to make sure the key messages stand out.

Technically I have a couple of comments that might be useful for improving this study and manuscript:

L45-46 This sentence does not have an obvious relationship with the context and needs further description.

L68-69 Then how well does the ASCAT soil moisture perform in Austria compared to other soil moisture products? Is it the best one? Have you had a chance to look at ESA CCI soil moisture product that blends a variety of passive and active microwave soil moisture products and seems to be more widely used.

L71 "The launch of the Sentinel-1 series provides observations at a high spatial resolution of 5x20 m". Can you clarify this further? Even though with higher spatial resolution using radars, does Sentinel-1 have sufficient spatial coverage to implement research like yours?

Table 1 This table can be moved to supplement materials.

L167-168 Were the parameters for different catchments different or the same? And how was the calibration scheme carried out? Were parameters for all catchments calibrated together or one by one?

L253 Why was this calibration period chosen? Typically, the calibration period should include historically wet, dry and average years.

L247-248: 0.3 seems to be a threshold value, is there any reason behind this?

L272-274 So can we pick out which weight allocation has the best performance for all three components. When do these components have the same weight? The conclusion would be easier to follow with fewer numbers.

L279-281 What does it mean with larger regional variability? This needs to be clarified further.

L283 "OSC tends to increase and the regional variability decreases for the variants involving SSC". Is there any reason behind this?

L289-293 The time coverage of soil moisture is nearly the same in the calibration and simulation periods (four water years), then why is the performance metric smaller in the calibration period than the simulation period? And why did the snow model perform better with fewer below zero temperature days?

L308-311 How did you derive this conclusion? Fig. 5 only shows the correlation between model performance and changing wQ, and readers cannot obtain information about changing attributes and their impacts on model performance.

L390-392 What does this stand for and what is the reason behind this? Given all the comments and issues indicated above, I recommend major to moderate revision prior to the acceptance of this manuscript into HESS.

---

## Referee Comment (RC2) · Anonymous Referee #2 · 24 Nov 2020

Review of **The value of ASCAT soil moisture and MODIS snow cover data for calibrating a conceptual hydrologic model** by Tong et al.

The study of Tong et al. focuses on using ASCAT soil moisture and MODIS snow cover, together with runoff for the calibration of the TUWmodel for 213 catchments in Austria. To do so, the study defines three calibration strategies for soil moisture and runoff, snow cover and runoff, and soil moisture, snow cover and runoff together. In addition, runoff is included with different weights for each calibration strategy. The authors conclude that the soil moisture data helps the soil moisture simulations, and the snow data the snow simulations.

Generally, the authors present a thorough analysis, and their conclusions are well supported by their data. Nevertheless, I need to raise some concerns, and I hope the authors can improve on these issues.

I think the manuscript could be made more concise and to-the-point. For example, twelve figures and 8 tables seems a bit much to me. Most figures and tables are also double, for the calibration and the validation, and the text becomes for that reason sometimes a bit repetitive. As these findings are rather similar, I would suggest that the authors focus more on one of the cases, for example the validation data, and move half of the figures to the Supplement. Some of the tables could be moved to the Supplement as well, as the figures already show the same data. I suggest as well to make Table 7 or 8 a similar figure as Figures 5 and 6, and remove the examples in Figures 11 and 12, that, in my view, do not add much compared to the full picture in the Tables 7 and 8. In addition, there are still relatively many unclear sentences and small issues with the figures., such as missing units.

More methodologically, the calibration on soil moisture is carried out for a much shorter period compared to the snow cover. At the same time, the authors conclude that the snow data are more efficient in improving snow simulations than the soil moisture data are in improving soil moisture simulations. Is it still a fair comparison? There is much more information in the snow data in this case.

To conclude, I generally like the approach and methodology, but some major improvements are needed. I hope the authors find my comments useful and I am looking forward to an improved version of the manuscript.

**Minor comments**
P1.L29. I think many more references could fit here, such as Kavetski et al. (2006), Wagener and Montanari (2011)
P5.L126. from digital elevation model → from a digital elevation model?
P5.L131. Except of → except for
P9.L233. Than the WQ variants. → What do you mean? To what are you comparing the red boxes in Fig.3? These are the WQ-variants already, correct?
P9.241. Particularly for WQ between 0.3 and 0.4. → How can I see this exactly? Please help the reader a bit.
P12.L294. signals .Accordingly → signals. Accordingly
P12.L291-295. Generally, it may also have to do with the depth that the remote sensing products "sees". The active rooting zones are much shallower in agricultural lands, whereas trees root much deeper.
P13.L303-304. The largest...only (wQ=1) →Do you mean for AP or in general?
P13.L304. Is larger than ...(SDGR). → I do not see this. You still discuss OSM in the second panel, correct?
P16.L228. The medians of the model parameters → of all the catchments, correct?

P16.L335-336. and adding snow and soil moisture is complementary → regarding the findings for the snow parameters you mean, correct? Please be specific.

P18.L375-376.has a median relative improvement about 4% to 25% lower → So you mean the difference between the red and blue line here? Seems more than 4 and 25% to me, if I look at it correctly.

P19.L386. WQ>0.6 → Not the other way around? WQ<0.6?

P24.L441. Previous studies → please add the references

P24.L441-445. If it is the same model, this is not very surprising and it doesn't seem to relate to the previous analyses as shown.

Sec.4.3. The title of this section relates also to what is discussed in the previous paragraph on Figure 8.

Fig3. Please add in the caption that the boxes represent the values from the different catchments. In addition, as WQ is a full calibration on Q only, I would here just show one boxplot. The three should be identical here, correct?

Fig.5. Is it correct that these come from the calibration SSM+SCM+runoff?

Fig.7. Please add the units of the parameters.

Fig.8. You could group the parameters here also for snow, soil moisture and runoff, maybe by just putting a thick line between them.

Table7,8. I would suggest to turn these Tables into similar figures as Figures 5 and 6.

**References**

Kavetski, D., Kuczera, G., Franks, S.W., 2006. Bayesian analysis of input uncertainty in hydrological modeling: 1. Theory. Water Resources Research 42. https://doi.org/10.1029/2005WR004368

Wagener, T., Montanari, A., 2011. Convergence of approaches toward reducing uncertainty in predictions in ungauged basins. Water Resources Research 47. https://doi.org/10.1029/2010WR009469

---

## Author Comment (AC1) · 14 Dec 2020

**Response Letter**

**The value of ASCAT soil moisture and MODIS snow cover data for calibrating a conceptual hydrologic model**

Rui Tong[1,2], Juraj Parajka[1,2], Andreas Salentinig[3], Isabella Pfeil[1,3], Jürgen Komma[2], Borbála Széles[1,2], Martin Kubáň[4], Peter Valent[2,4], Mariette Vreugdenhil[3], Wolfgang Wagner[1,3] and Günter Blöschl[1,2]

[1]Centre for Water Resource Systems, TU Wien, Vienna, 1040, Austria
[2]Institute of Hydraulic Engineering and Water Resources Management, TU Wien, Vienna, 1040, Austria
[3]Department of Geodesy and Geoinformation, TU Wien, Vienna, 1040, Austria
[4]Department of Land and Water Resources Management, Slovak University of Technology, Bratislava, 810 05, Slovakia

In the following document, we reproduce all the comments of the Referees in italic characters followed by our responses.

**Response to referee #1**

*The authors calibrated a conceptual hydrological model by using ASCAT soil moisture and MODIS snow cover data jointly or separately, and improvements of related variants have been achieved in varying degrees. The efficiency improvement was also analyzed under different scenarios and catchment attributes. Overall, the results seem convincing and the study is valuable for related research. However, there are several issues that still exist and need to be clarified further as indicated in the following.*

> We want to thank the reviewer for her/his positive, helpful and insightful comments on the manuscript.

*First, the manuscript needs further editorial work to improve the paragraph structure and some vague expressions. A single sentence definitely cannot be a paragraph (e.g., line 286), and a paragraph should not be too long or too short. In addition, please pay attention to vague expressions in this manuscript, such as line 84 " to compare the multiple objective calibration to soil moisture and runoff to three different calibration variants", which is really confusing. There are other similar sentences, so I hope the authors make a thorough change to improve the clarity of the manuscript.*

> Thank you for this comment. In response to this comment, we have checked the manuscript and tried to improve the clarity of presentation and length of paragraphs. We made numerous edits to improve the clarity of presentation. For example, the break line on l. 286 has been removed, or long sentences (asm e.g. on l.84) have been simplified.

*Another major issue in this manuscript is that conclusions in the Results section can be presented in a more straightforward way. At its current form, the conclusions are over detailed and have too many numbers, which do not have much value. It is hard for the readers to get the key messages from the authors. Furthermore, the figures and tables contain too much information (e.g., Fig5-6, Table 7-8), also leading to the difficulty in deriving the key information. So please make more concise and clearer conclusions, and improve the presentation of all figures and tables to make sure the key messages stand out.*

In response to this comment, we have reduced the number of variables showed in Figures 5 and 6. The variables which do not have significant correlation have been moved to a new Figure, which was moved to the Supplement. We have also moved the Tables 7 and 8 to the Supplement.

*Technically I have a couple of comments that might be useful for improving this study and manuscript:*

*L45-46 This sentence does not have an obvious relationship with the context and needs further description.*

> The study of Nijzink et al. (2018) belongs to studies evaluating the value of the combination of different products for constraining hydrologic models. This study shows that the combination of different soil moisture satellite products can help to constrain not only soil but also snow model parameters. In response to this comment, we have revised the sentence as follows: "For example, Nijzink et al. (2018) demonstrated that constraining hydrologic models profited from an increased number of data sources. Interestingly, the use of different soil moisture products had a positive impact on the identifiability of not only soil but also snow model parameters."

*L68-69 Then how well does the ASCAT soil moisture perform in Austria compared to other soil moisture products? Is it the best one? Have you had a chance to look at ESA CCI soil moisture product that blends a variety of passive and active microwave soil moisture products and seems to be more widely used.*

> The ASCAT product used in this research is the same product which is used as the active product in CCI. The study of Dorigo et al. (2017) demonstrated the quality of the active CCI product over temperate climates such as Austria. This is demonstrated in the blending weights used for the combined product, which are based on triple colocation analysis. Over Austria, the active product has the largest weight, close to 1 (Figure 2). Considering that the spatial sampling of the CCI dataset is 0.25deg, whereas the ASCAT product has a sampling of 12.5km we chose to use the improved ASCAT product in this research. In addition, we have improved the algorithm of the ASCAT product to perform better over Austria. Among other algorithm changes, improved vegetation parameters have been applied as described in Pfeil et al. 2018. In this paper, the authors also showed that the ASCAT, AMSR-2 and SMAP soil moisture products perform similarly well over a temperate climate agricultural catchment.

> In response to this comment, we added below in the discussion:

"The ASCAT product used in this research is the same product which is used as the active product in the ESA-CCI. The study of Dorigo et al. (2017) demonstrated the quality of the active ESA-CCI product over temperate climates such as Austria. Considering that the spatial sampling of the ESA-CCI dataset is 0.25 degree, whereas the ASCAT product has a sampling of 12.5 km, the ASCAT product was chosen to be applied in this study. In addition, the algorithm of the ASCAT product which improved vegetation parameters performed better over Austria (Pfeil et al., 2018)."

*L71 "The launch of the Sentinel-1 series provides observations at a high spatial resolution of x20 m". Can you clarify this further? Even though with higher spatial resolution using radars, does Sentinel-1 have sufficient spatial coverage to implement research like yours?*

Sentinel-1 has global coverage, but because of the acquisition strategy, most observations are over Europe, where a temporal resolution of 1.5-4 days can be achieved. Globally the temporal resolution decreases, but the high-resolution SWI can still be obtained as the directional resampling parameters can be calculated reliably with the number of Sentinel-1 observations available.

*Table 1 This table can be moved to supplement materials.*

In response to this comment, we have moved this table to the supplementary.

*L167-168 Were the parameters for different catchments different or the same? And how was the calibration scheme carried out? Were parameters for all catchments calibrated together or one by one?*

The parameters for different catchments are different. The parameter calibration scheme was described in section 3.2. In response to this comment, we added the sentence in L211 "The procedure of model parameters calibration is carried out for each calibration variant and each catchment independently."

*L253 Why was this calibration period chosen? Typically, the calibration period should include historically wet, dry and average years.*

The idea was to split the period with available (overlapped) snow, soil moisture and runoff observations into two periods. The calibration period contains the year 2005, which is flood rich; also the validation period contains the year 2013, which is also flood rich.

*L247-248: 0.3 seems to be a threshold value, is there any reason behind this?*

This is a good question. The results are based on a large sample of catchments, so there is not a simple answer. We plan to investigate further the factors controlling the change in the efficiency in future analyses.

*L272-274 So can we pick out which weight allocation has the best performance for all three components. When do these components have the same weight? The conclusion would be easier to follow with fewer numbers.*

We believe that plotting individual efficiency measures will allow a more robust interpretation of results than merging (and weighting) them together, so we prefer not to change this Figure and leave it in its current form.

*L279-281 What does it mean with larger regional variability? This needs to be clarified further.*

In Line249-250, we illustrated the regional variability is the variability between catchments.

*L283 "OSC tends to increase and the regional variability decreases for the variants involving SSC". Is there any reason behind this?*

These results indicate some compensation effects of a degree-day approach when simulating both, the snow cover dynamics and snowmelt runoff. Increasing the weight to runoff (i.e. to mimic more snowmelt runoff by the model) tends, in some catchments, to be at the expense of the accuracy of snow cover mapping. There can be different factors affecting the snow model efficiency, including spatial estimation of the input, uncertainty of satellite data or simplification of the snowmelt process by the degree-day approach. The detailed analysis of such reasons goes, however, beyond the main scope of the paper.

*L289-293 The time coverage of soil moisture is nearly the same in the calibration and simulation periods (four water years), then why is the performance metric smaller in the calibration period than the simulation period? And why did the snow model perform better with fewer below zero temperature days?*

The reason for the first question is likely to be caused by the length for the validation is 11% (median) longer than the calibration. Secondly, but not able to be confirmed, since 2012, the second ASCAT satellite was onboard, which might improve the quality of soil moisture data.

The second question is related to the no snow condition, which is easier to be simulated for the model.

*L308-311 How did you derive this conclusion? Fig. 5 only shows the correlation between model performance and changing wQ, and readers cannot obtain information about changing attributes and their impacts on model performance.*

This part refers to a comparison between Fig 5, 6 and Figs S1-S4 in the Supplement which shows that the correlation between model efficiency and catchment attributes and its change with the runoff weight is similar for all calibration variants. In response to this comment, we have added following explanation: "It is obvious that for runoff weight $w_Q$>=0.4 for $O_Q$, $w_Q$>=0.0 for $O_{SM}$, and $w_Q$>=0.1 for $O_{SC}$, the correlations between model efficiency and catchment characteristics are similar to that for the runoff only calibration."

*L390-392 What does this stand for and what is the reason behind this?*

This part refers to black lines in Fig. 9. In response to this comment, we have rephrased the sentence to:

"While the inclusion of soil moisture data in the calibration mainly improves the soil moisture simulations and the inclusion of snow data improves the snow simulations, the use of both in model calibration improves both soil moisture and snow simulations to a similar extent."

Reference

Dorigo, Wouter, Wolfgang Wagner, Clement Albergel, Franziska Albrecht, Gianpaolo Balsamo, Luca Brocca, Daniel Chung, Martin Ertl, Matthias Forkel, and Alexander Gruber. 2017. "ESA CCI Soil Moisture for Improved Earth System Understanding: State-of-the Art and Future Directions." Remote Sensing of Environment 203: 185–215.

Nijzink, R. C., Almeida, S., Pechlivanidis, I. G., Capell, R., Gustafssons, D., Arheimer, B., Parajka, J., Freer, J., Han, D., Wagener, T., van Nooijen, R. R. P., Savenije, H. H. G., and Hrachowitz, M.: Constraining Conceptual Hydrological Models With Multiple Information Sources, Water Resources Research, 54, 8332-8362, https://doi.org/10.1029/2017wr021895, 2018.

Pfeil, I., M. Vreugdenhil, S. Hahn, W. Wagner, P. Strauss, and G. Blöschl. 2018. "Improving the Seasonal Representation of ASCAT Soil Moisture and Vegetation Dynamics in a Temperate Climate." Remote Sensing 10 (11). https://doi.org/10.3390/rs10111788.

---

## Author Comment (AC2) · 14 Dec 2020

**Response Letter**

**The value of ASCAT soil moisture and MODIS snow cover data for calibrating a conceptual hydrologic model**

Rui Tong[1,2], Juraj Parajka[1,2], Andreas Salentinig[3], Isabella Pfeil[1,3], Jürgen Komma[2], Borbála Széles[1,2], Martin Kubáň[4], Peter Valent[2,4], Mariette Vreugdenhil[3], Wolfgang Wagner[1,3] and Günter Blöschl[1,2]

[1]Centre for Water Resource Systems, TU Wien, Vienna, 1040, Austria
[2]Institute of Hydraulic Engineering and Water Resources Management, TU Wien, Vienna, 1040, Austria
[3]Department of Geodesy and Geoinformation, TU Wien, Vienna, 1040, Austria
[4]Department of Land and Water Resources Management, Slovak University of Technology, Bratislava, 810 05, Slovakia

In the following document, we reproduce all the comments of the Referees in italic characters followed by our responses.

**Response to referee #2**

*The study of Tong et al. focuses on using ASCAT soil moisture and MODIS snow cover, together with runoff for the calibration of the TUWmodel for 213 catchments in Austria. To do so, the study defines three calibration strategies for soil moisture and runoff, snow cover and runoff, and soil moisture, snow cover and runoff together. In addition, runoff is included with different weights for each calibration strategy. The authors conclude that the soil moisture data helps the soil moisture simulations, and the snow data the snow simulations.*

*Generally, the authors present a thorough analysis, and their conclusions are well supported by their data. Nevertheless, I need to raise some concerns, and I hope the authors can improve on these issues.*

*I think the manuscript could be made more concise and to-the-point. For example, twelve figures and 8 tables seems a bit much to me. Most figures and tables are also double, for the calibration and the validation, and the text becomes for that reason sometimes a bit repetitive. As these findings are rather similar, I would suggest that the authors focus more on one of the cases, for example the validation data, and move half of the figures to the Supplement. Some of the tables could be moved to the Supplement as well, as the figures already show the same data. I suggest as well to make Table 7 or 8 a similar figure as Figures 5 and 6, and remove the examples in Figures 11 and 12, that, in my view, do not add much compared to the full picture in the Tables 7 and 8. In addition, there are still relatively many unclear sentences and small issues with the figures., such as missing units.*

*In response to this comment, we have simplified the Figs 5 and 6 and moved a part of them to the Supplement. We have also moved Table 7 and 8 and Figs 11 and 12 to the Supplement, as suggested by the reviewer.*

*More methodologically, the calibration on soil moisture is carried out for a much shorter period compared to the snow cover. At the same time, the authors conclude that the snow data are more efficient in improving snow simulations than the soil moisture data are in improving soil moisture simulations. Is it still a fair comparison? There is much more information in the snow data in this case.*

*It is true that in this study, the observation records of soil moisture are less than for snow cover. In a follow up study (Kuban et al, in preparation) we test a longer calibration period (2005-2014) and the soil moisture efficiencies are very similar as found in this study. Therefore, we believe that the availability of soil moisture and selection of the calibration/validation periods here does not have an impact on the interpretation of results. We plan also to investigate further the impact and inter-relation of snow and soil moisture data in the next study.*

*To conclude, I generally like the approach and methodology, but some major improvements are needed. I hope the authors find my comments useful and I am looking forward to an improved version of the manuscript.*

> We want to thank the reviewer for her/his positive, helpful and insightful comments on the manuscript.

*Minor comments*

*P1.L29. I think many more references could fit here, such as Kavetski et al. (2006), Wagener and Montanari (2011)*

> References added.

*P5.L126. from digital elevation model → from a digital elevation model?*

> Revised as suggested.

*P5.L131. Except of → except for*

> Revised as suggested.

*P9.L233. Than the WQ variants. → What do you mean? To what are you comparing the red boxes in Fig.3? These are the WQ-variants already, correct?*

> Yes, it is a typo, thank you for pointing this. We have rephrased the sentence as follows:" The correlation between ASCAT and simulated soil moisture (Figure 3, centre) has a much larger regional variability (i.e. variability between catchments) than the variability of OQ (Fig.3, top panel) for all wQ. ".

*P9.241. Particularly for WQ between 0.3 and 0.4. → How can I see this exactly? Please help the reader a bit.*

> More quantitatively is this relative improvement and difference between calibration variants evaluated in Figure9. From this evaluation it is clear that the wQ=0.3 has the largest relative improvement if merging all three performance measures.

*P12.L294. signals .Accordingly → signals. Accordingly*

Revised as suggested

*P12.L291-295. Generally, it may also have to do with the depth that the remote sensing products "sees". The active rooting zones are much shallower in agricultural lands, whereas trees root much deeper.*

Thank you very much for your suggestion. We added "The active rooting zones are much shallower in agricultural lands, whereas trees root much deeper. Hence the satellite soil moisture data used in this study which monitored for the top 100 cm soil layer may fit the soil moisture for arable land better." to explain this point.

*P13.L303-304. The largest...only (wQ=1) →Do you mean for AP or in general?*

It is in general. In response we modify this sentence to "The largest negative correlation of soil moisture efficiency and attributions is found for calibration to runoff only ($w\_Q=1$) in general ..."

*P13.L304. Is larger than ...(SDGR). → I do not see this. You still discuss OSM in the second panel, correct?*

True, there was mistakes. In response we modify it to "…is larger than 0.7 for MELE, catchment elevation range (ER), and standard deviation of MAT and mean daily potential global radiation (SDGR)."

*P16.L328. The medians of the model parameters → of all the catchments, correct?*

Yes, we added "for all catchments".

*P16.L335-336. and adding snow and soil moisture is complementary → regarding the findings for the snow parameters you mean, correct? Please be specific.*

We modified it to "This suggests that adding soil moisture satellite data in model calibration affects the soil-related parameters strongly and adding snow and soil moisture satellite data is complementary for influencing both snow and soil moisture related parameters."

*P18.L375-376.has a median relative improvement about 4% to 25% lower → So you mean the difference between the red and blue line here? Seems more than 4 and 25% to me, if I look at it correctly.*

Thank you for the remind. It should be 3%-35%.

*P19.L386. WQ>0.6 → Not the other way around? WQ<0.6?*

Yes, less than.

*P24.L441. Previous studies → please add the references*

We added the references as below: Parajka et al., 2008, 2009; Sleziak et al., 2018;

*P24.L441-445. If it is the same model, this is not very surprising and it doesn't seem to relate to the previous analyses as shown.*

Indeed, it is not surprising, but we wanted to provide a link to previous studies performed by the same model, in the same region.

*Sec.4.3. The title of this section relates also to what is discussed in the previous paragraph on Figure 8.*

The idea of section 4.3 is to compare the model efficiencies. In response to this comment we have revised the title to:

Comparison of multiple objective and runoff only calibration efficiencies

*Fig3. Please add in the caption that the boxes represent the values from the different catchments. In addition, as WQ is a full calibration on Q only, I would here just show one boxplot. The three should be identical here, correct?*

Revised as suggested.

*Fig.5. Is it correct that these come from the calibration SSM+SCM+runoff?*

Yes, in response we added "SSM+SCM+runoff" in the caption.

*Fig.7. Please add the units of the parameters.*

Units added.

*Fig.8. You could group the parameters here also for snow, soil moisture and runoff, maybe by just putting a thick line between them.*

Revised as suggested.

*Table7,8. I would suggest to turn these Tables into similar figures as Figures 5 and 6.*

Taking into account the suggestions of reviewer #1 and comments related to the number of figures, we have moved these Tables into the Supplement.

---

## Referee Report (RR1)

Review of **The value of ASCAT soil moisture and MODIS snow cover data for calibrating a conceptual hydrologic model** by Tong et al.

The revised manuscript of Tong et al, that deals with different calibration strategies based on runoff, soil moisture and snow cover, shows many improvements compared to the previous version. I am happy the authors looked at their tables critically and moved a substantial amount of figures and tables to the supplement.

However, I would like to clarify one of my comments in the previous round, as I think the authors misunderstood here and moved it a bit to the other extreme. The authors moved Tables 7 and 8 (now S2 and S3) to the supplement, but I found these actually interesting and suggested to use one of these tables to make a similar figure as Figures 5,6 or 8. Now, the paragraph on page 20 is solely about the supplementary material, but some information on that in the main manuscript would be nice. I was also mainly referring to Tables 4 and 5 (in the new version of the manuscript) as these show data that is also displayed in the figures, that are therefore redundant and act more as background information which is more suitable for the Supplement. These are also more suggestions from my side that, in my view, could improve the manuscript, but I leave this a bit to the authors.

I have the feeling the text also improved a lot (I believe the tracked changes version does not contain all changes), but think there are still some unclarities in the manuscript. For example, referencing the subfigures with labels from a-z (please see also the HESS guidelines for subfigures) and referring to the exact subfigures would probably help a lot already in discussing the different panels. See also my minor comments for more specific issues.

I believe these comments are rather minor, and mainly textual. I hope the authors find them useful again, and look forward to a final version of the manuscript.

**Minor comments**
P1.L27. Model parameters estimation methods → model parameter estimation methods
P2.L41. Models has been improved → models improved.
P2.L51. Volumes. (Zhang et al., 2009). → Volumes (Zhang et al., 2009).
P2.L53. Which efficiency?
P3.L65. (Kim et al. 2020) but → (Kim et al. 2020), but
P3.L66. Nonetheless there → Nonetheless, there
P3.L83-84. to compare ... calibration variants → this sounds a bit confusing to me, could you rephrase?
P3.L86 which → that
P4.L95. data product.. → data product.
P4.L119-120. Pixels classified...or snow free → and vice versa, correct?
P5.L141 maps which → maps, which
P5.L144. Why are these numbers different? Are the 208 a subset of the 213? Before, only 213 is mentioned (P4.L122).
P6.L158. Storage coefficient → storage coefficients
P8.L202. Why should SWE be bigger than 10mm?
P11.L256-257. The SSC … Wq <0.3. → You mean compared to the calibration or the other variant?
P11.L273 proportion of day → proportion of days?
P13.L293. Alpine catchment → Alpine catchments?
P13.L301. Are the soil moisture data used here actually the same as for the calibration? So remotely sensed soil moisture?
P15.L336-337. In contrast the → In contrast, the

P15.L338. Tr, Ts, Tm → Figure 7 only has meltT, is that Tm?

P17-18.L377-388. I would suggest to give the subpanels in Figure 9 letters a-f and add these references in the text. Please also be specific which model efficiency you discuss ($O_q$, $O_{SM}$, $O_{SC}$). This paragraph is currently a bit confusing occasionally.

P20.L427. There is no Figure 11 anymore.

P20. This page is now only about supplementary material. Why not add one figure with the p-values?

---

## Author Response (AR2)

**Response Letter**

**The value of ASCAT soil moisture and MODIS snow cover data for calibrating a conceptual hydrologic model**

Rui Tong[1,2], Juraj Parajka[1,2], Andreas Salentinig[3], Isabella Pfeil[1,3], Jürgen Komma[2], Borbála Széles[1,2], Martin Kubáň[4], Peter Valent[2,4], Mariette Vreugdenhil[3], Wolfgang Wagner[1,3] and Günter Blöschl[1,2]

[1]Centre for Water Resource Systems, TU Wien, Vienna, 1040, Austria
[2]Institute of Hydraulic Engineering and Water Resources Management, TU Wien, Vienna, 1040, Austria
[3]Department of Geodesy and Geoinformation, TU Wien, Vienna, 1040, Austria
[4]Department of Land and Water Resources Management, Slovak University of Technology, Bratislava, 810 05, Slovakia

In the following document, we reproduce all the comments of the Referees in italic characters followed by our responses.

**First round**

**Response to referee #1**

*The authors calibrated a conceptual hydrological model by using ASCAT soil moisture and MODIS snow cover data jointly or separately, and improvements of related variants have been achieved in varying degrees. The efficiency improvement was also analyzed under different scenarios and catchment attributes. Overall, the results seem convincing and the study is valuable for related research. However, there are several issues that still exist and need to be clarified further as indicated in the following.*

> We want to thank the reviewer for her/his positive, helpful and insightful comments on the manuscript.

*First, the manuscript needs further editorial work to improve the paragraph structure and some vague expressions. A single sentence definitely cannot be a paragraph (e.g., line 286), and a paragraph should not be too long or too short. In addition, please pay attention to vague expressions in this manuscript, such as line 84 " to compare the multiple objective calibration to soil moisture and runoff to three different calibration variants", which is really confusing. There are other similar sentences, so I hope the authors make a thorough change to improve the clarity of the manuscript.*

> Thank you for this comment. In response to this comment, we have checked the manuscript and tried to improve the clarity of presentation and length of paragraphs. We made numerous edits to improve the clarity of presentation. For example, the break line on l. 286 has been removed, or long sentences (asm e.g. on l.84) have been simplified.

*Another major issue in this manuscript is that conclusions in the Results section can be presented in a more straightforward way. At its current form, the conclusions are over detailed and have too many numbers, which do not have much value. It is hard for the readers to get the key messages from the authors. Furthermore, the figures and tables contain too much information (e.g., Fig5-6, Table 7-8), also leading to the difficulty in deriving the key information. So please make more concise and clearer conclusions, and improve the presentation of all figures and tables to make sure the key messages stand out.*

In response to this comment, we have reduced the number of variables showed in Figures 5 and 6. The variables which do not have significant correlation have been moved to a new Figure, which was moved to the Supplement. We have also moved the Tables 7 and 8 to the Supplement.

*Technically I have a couple of comments that might be useful for improving this study and manuscript:*

*L45-46 This sentence does not have an obvious relationship with the context and needs further description.*

> The study of Nijzink et al. (2018) belongs to studies evaluating the value of the combination of different products for constraining hydrologic models. This study shows that the combination of different soil moisture satellite products can help to constrain not only soil but also snow model parameters. In response to this comment, we have revised the sentence as follows: "For example, Nijzink et al. (2018) demonstrated that constraining hydrologic models profited from an increased number of data sources. Interestingly, the use of different soil moisture products had a positive impact on the identifiability of not only soil but also snow model parameters."

*L68-69 Then how well does the ASCAT soil moisture perform in Austria compared to other soil moisture products? Is it the best one? Have you had a chance to look at ESA CCI soil moisture product that blends a variety of passive and active microwave soil moisture products and seems to be more widely used.*

> The ASCAT product used in this research is the same product which is used as the active product in CCI. The study of Dorigo et al. (2017) demonstrated the quality of the active CCI product over temperate climates such as Austria. This is demonstrated in the blending weights used for the combined product, which are based on triple colocation analysis. Over Austria, the active product has the largest weight, close to 1 (Figure 2). Considering that the spatial sampling of the CCI dataset is 0.25deg, whereas the ASCAT product has a sampling of 12.5km we chose to use the improved ASCAT product in this research. In addition, we have improved the algorithm of the ASCAT product to perform better over Austria. Among other algorithm changes, improved vegetation parameters have been applied as described in Pfeil et al. 2018. In this paper, the authors also showed that the ASCAT, AMSR-2 and SMAP soil moisture products perform similarly well over a temperate climate agricultural catchment.

In response to this comment, we added below in the discussion:

"The ASCAT product used in this research is the same product which is used as the active product in the ESA-CCI. The study of Dorigo et al. (2017) demonstrated the quality of the active ESA-CCI product over temperate climates such as Austria. Considering that the spatial sampling of the ESA-CCI dataset is 0.25 degree, whereas the ASCAT product has a sampling of 12.5 km, the ASCAT product was chosen to be applied in this study. In addition, the algorithm of the ASCAT product which improved vegetation parameters performed better over Austria (Pfeil et al., 2018)."

*L71 "The launch of the Sentinel-1 series provides observations at a high spatial resolution of x20 m". Can you clarify this further? Even though with higher spatial resolution using radars, does Sentinel-1 have sufficient spatial coverage to implement research like yours?*

Sentinel-1 has global coverage, but because of the acquisition strategy, most observations are over Europe, where a temporal resolution of 1.5-4 days can be achieved. Globally the temporal resolution decreases, but the high-resolution SWI can still be obtained as the directional resampling parameters can be calculated reliably with the number of Sentinel-1 observations available.

*Table 1 This table can be moved to supplement materials.*

In response to this comment, we have moved this table to the supplementary.

*L167-168 Were the parameters for different catchments different or the same? And how was the calibration scheme carried out? Were parameters for all catchments calibrated together or one by one?*

The parameters for different catchments are different. The parameter calibration scheme was described in section 3.2. In response to this comment, we added the sentence in L211 "The procedure of model parameters calibration is carried out for each calibration variant and each catchment independently."

*L253 Why was this calibration period chosen? Typically, the calibration period should include historically wet, dry and average years.*

The idea was to split the period with available (overlapped) snow, soil moisture and runoff observations into two periods. The calibration period contains the year 2005, which is flood rich; also the validation period contains the year 2013, which is also flood rich.

*L247-248: 0.3 seems to be a threshold value, is there any reason behind this?*

This is a good question. The results are based on a large sample of catchments, so there is not a simple answer. We plan to investigate further the factors controlling the change in the efficiency in future analyses.

*L272-274 So can we pick out which weight allocation has the best performance for all three components. When do these components have the same weight? The conclusion would be easier to follow with fewer numbers.*

We believe that plotting individual efficiency measures will allow a more robust interpretation of results than merging (and weighting) them together, so we prefer not to change this Figure and leave it in its current form.

*L279-281 What does it mean with larger regional variability? This needs to be clarified further.*

In Line249-250, we illustrated the regional variability is the variability between catchments.

*L283 "OSC tends to increase and the regional variability decreases for the variants involving SSC". Is there any reason behind this?*

These results indicate some compensation effects of a degree-day approach when simulating both, the snow cover dynamics and snowmelt runoff. Increasing the weight to runoff (i.e. to mimic more snowmelt runoff by the model) tends, in some catchments, to be at the expense of the accuracy of snow cover mapping. There can be different factors affecting the snow model efficiency, including spatial estimation of the input, uncertainty of satellite data or simplification of the snowmelt process by the degree-day approach. The detailed analysis of such reasons goes, however, beyond the main scope of the paper.

*L289-293 The time coverage of soil moisture is nearly the same in the calibration and simulation periods (four water years), then why is the performance metric smaller in the calibration period than the simulation period? And why did the snow model perform better with fewer below zero temperature days?*

The reason for the first question is likely to be caused by the length for the validation is 11% (median) longer than the calibration. Secondly, but not able to be confirmed, since 2012, the second ASCAT satellite was onboard, which might improve the quality of soil moisture data.

The second question is related to the no snow condition, which is easier to be simulated for the model.

*L308-311 How did you derive this conclusion? Fig. 5 only shows the correlation between model performance and changing wQ, and readers cannot obtain information about changing attributes and their impacts on model performance.*

This part refers to a comparison between Fig 5, 6 and Figs S1-S4 in the Supplement which shows that the correlation between model efficiency and catchment attributes and its change with the runoff weight is similar for all calibration variants. In response to this comment, we have added following explanation: "It is obvious that for runoff weight $w_Q>=0.4$ for $O_Q$, $w_Q>=0.0$ for $O_{SM}$, and $w_Q>=0.1$ for $O_{SC}$, the correlations between model efficiency and catchment characteristics are similar to that for the runoff only calibration."

*L390-392 What does this stand for and what is the reason behind this?*

This part refers to black lines in Fig. 9. In response to this comment, we have rephrased the sentence to:

"While the inclusion of soil moisture data in the calibration mainly improves the soil moisture simulations and the inclusion of snow data improves the snow simulations, the use of both in model calibration improves both soil moisture and snow simulations to a similar extent."

Reference

Dorigo, Wouter, Wolfgang Wagner, Clement Albergel, Franziska Albrecht, Gianpaolo Balsamo, Luca Brocca, Daniel Chung, Martin Ertl, Matthias Forkel, and Alexander Gruber. 2017. "ESA CCI Soil Moisture for Improved Earth System Understanding: State-of-the Art and Future Directions." Remote Sensing of Environment 203: 185–215.

Nijzink, R. C., Almeida, S., Pechlivanidis, I. G., Capell, R., Gustafssons, D., Arheimer, B., Parajka, J., Freer, J., Han, D., Wagener, T., van Nooijen, R. R. P., Savenije, H. H. G., and Hrachowitz, M.: Constraining Conceptual Hydrological Models With Multiple Information Sources, Water Resources Research, 54, 8332-8362, https://doi.org/10.1029/2017wr021895, 2018.

Pfeil, I., M. Vreugdenhil, S. Hahn, W. Wagner, P. Strauss, and G. Blöschl. 2018. "Improving the Seasonal Representation of ASCAT Soil Moisture and Vegetation Dynamics in a Temperate Climate." Remote Sensing 10 (11). https://doi.org/10.3390/rs10111788.

**Response to referee #2**

*The study of Tong et al. focuses on using ASCAT soil moisture and MODIS snow cover, together with runoff for the calibration of the TUWmodel for 213 catchments in Austria. To do so, the study defines three calibration strategies for soil moisture and runoff, snow cover and runoff, and soil moisture, snow cover and runoff together. In addition, runoff is included with different weights for each calibration strategy. The authors conclude that the soil moisture data helps the soil moisture simulations, and the snow data the snow simulations.*

*Generally, the authors present a thorough analysis, and their conclusions are well supported by their data. Nevertheless, I need to raise some concerns, and I hope the authors can improve on these issues.*

*I think the manuscript could be made more concise and to-the-point. For example, twelve figures and 8 tables seems a bit much to me. Most figures and tables are also double, for the calibration and the validation, and the text becomes for that reason sometimes a bit repetitive. As these findings are rather similar, I would suggest that the authors focus more on one of the cases, for example the validation data, and move half of the figures to the Supplement. Some of the tables could be moved to the Supplement as well, as the figures already show the same data. I suggest as well to make Table 7 or 8 a similar figure as Figures 5 and 6, and remove the examples in Figures 11 and 12, that, in my view, do not add much compared to the full picture in the Tables 7 and 8. In addition, there are still relatively many unclear sentences and small issues with the figures., such as missing units.*

> In response to this comment, we have simplified the Figs 5 and 6 and moved a part of them to the Supplement. We have also moved Table 7 and 8 and Figs 11 and 12 to the Supplement, as suggested by the reviewer.

*More methodologically, the calibration on soil moisture is carried out for a much shorter period compared to the snow cover. At the same time, the authors conclude that the snow data are more efficient in improving snow simulations than the soil moisture data are in improving soil moisture*

*simulations. Is it still a fair comparison? There is much more information in the snow data in this case.*

It is true that in this study, the observation records of soil moisture are less than for snow cover. In a follow up study (Kuban et al, in preparation) we test a longer calibration period (2005-2014) and the soil moisture efficiencies are very similar as found in this study. Therefore, we believe that the availability of soil moisture and selection of the calibration/validation periods here does not have an impact on the interpretation of results. We plan also to investigate further the impact and inter-relation of snow and soil moisture data in the next study.

*To conclude, I generally like the approach and methodology, but some major improvements are needed. I hope the authors find my comments useful and I am looking forward to an improved version of the manuscript.*

We want to thank the reviewer for her/his positive, helpful and insightful comments on the manuscript.

*Minor comments*

*P1.L29. I think many more references could fit here, such as Kavetski et al. (2006), Wagener and Montanari (2011)*

References added.

*P5.L126. from digital elevation model → from a digital elevation model?*

Revised as suggested.

*P5.L131. Except of → except for*

Revised as suggested.

*P9.L233. Than the WQ variants. → What do you mean? To what are you comparing the red boxes in Fig.3? These are the WQ-variants already, correct?*

Yes, it is a typo, thank you for pointing this. We have rephrased the sentence as follows:" The correlation between ASCAT and simulated soil moisture (Figure 3, centre) has a much larger regional variability (i.e. variability between catchments) than the variability of OQ (Fig.3, top panel) for all wQ. ".

*P9.241. Particularly for WQ between 0.3 and 0.4. → How can I see this exactly? Please help the reader a bit.*

More quantitatively is this relative improvement and difference between calibration variants evaluated in Figure9. From this evaluation it is clear that the wQ=0.3 has the largest relative improvement if merging all three performance measures.

*P12.L294. signals .Accordingly → signals. Accordingly*

Revised as suggested

*P12.L291-295. Generally, it may also have to do with the depth that the remote sensing products "sees". The active rooting zones are much shallower in agricultural lands, whereas trees root much deeper.*

Thank you very much for your suggestion. We added "The active rooting zones are much shallower in agricultural lands, whereas trees root much deeper. Hence the satellite soil moisture data used in this study which monitored for the top 100 cm soil layer may fit the soil moisture for arable land better." to explain this point.

*P13.L303-304. The largest...only (wQ=1) →Do you mean for AP or in general?*

It is in general. In response we modify this sentence to "The largest negative correlation of soil moisture efficiency and attributions is found for calibration to runoff only (w_Q=1) in general ..."

*P13.L304. Is larger than ...(SDGR). → I do not see this. You still discuss OSM in the second panel, correct?*

True, there was mistakes. In response we modify it to "…is larger than 0.7 for MELE, catchment elevation range (ER), and standard deviation of MAT and mean daily potential global radiation (SDGR)."

*P16.L328. The medians of the model parameters → of all the catchments, correct?*

Yes, we added "for all catchments".

*P16.L335-336. and adding snow and soil moisture is complementary → regarding the findings for the snow parameters you mean, correct? Please be specific.*

We modified it to "This suggests that adding soil moisture satellite data in model calibration affects the soil-related parameters strongly and adding snow and soil moisture satellite data is complementary for influencing both snow and soil moisture related parameters."

*P18.L375-376.has a median relative improvement about 4% to 25% lower → So you mean the difference between the red and blue line here? Seems more than 4 and 25% to me, if I look at it correctly.*

Thank you for the remind. It should be 3%-35%.

*P19.L386. WQ>0.6 → Not the other way around? WQ<0.6?*

Yes, less than.

*P24.L441. Previous studies → please add the references*

We added the references as below: Parajka et al., 2008, 2009; Sleziak et al., 2018;

*P24.L441-445. If it is the same model, this is not very surprising and it doesn't seem to relate to the previous analyses as shown.*

Indeed, it is not surprising, but we wanted to provide a link to previous studies performed by the same model, in the same region.

*Sec.4.3. The title of this section relates also to what is discussed in the previous paragraph on Figure 8.*

The idea of section 4.3 is to compare the model efficiencies. In response to this comment we have revised the title to:

Comparison of multiple objective and runoff only calibration efficiencies

*Fig3. Please add in the caption that the boxes represent the values from the different catchments. In addition, as WQ is a full calibration on Q only, I would here just show one boxplot. The three should be identical here, correct?*

Revised as suggested.

*Fig.5. Is it correct that these come from the calibration SSM+SCM+runoff?*

Yes, in response we added "SSM+SCM+runoff" in the caption.

*Fig.7. Please add the units of the parameters.*

Units added.

*Fig.8. You could group the parameters here also for snow, soil moisture and runoff, maybe by just putting a thick line between them.*

Revised as suggested.

*Table7,8. I would suggest to turn these Tables into similar figures as Figures 5 and 6.*

Taking into account the suggestions of reviewer #1 and comments related to the number of figures, we have moved these Tables into the Supplement.

**Second round**

**Response to referee #1**

*Review of The value of ASCAT soil moisture and MODIS snow cover data for calibrating a conceptual hydrologic model by Tong et al. The revised manuscript of Tong et al, that deals with different calibration strategies based on runoff, soil moisture and snow cover, shows many improvements compared to the previous version. I am happy the authors looked at their tables critically and moved a substantial amount of figures and tables to the supplement.*

We want to thank the reviewer for her/his positive and helpful comments.

*However, I would like to clarify one of my comments in the previous round, as I think the authors*

*misunderstood here and moved it a bit to the other extreme. The authors moved Tables 7 and 8 (now S2 and S3) to the supplement, but I found these actually interesting and suggested to use one of these tables to make a similar figure as Figures 5,6 or 8. Now, the paragraph on page 20 is solely about the supplementary material, but some information on that in the main manuscript would be nice. I was also mainly referring to Tables 4 and 5 (in the new version of the manuscript) as these show data that is also displayed in the figures, that are therefore redundant and act more as background information which is more suitable for the Supplement. These are*

*also more suggestions from my side that, in my view, could improve the manuscript, but I leave this a bit to the authors.*

In response to this comment, we have moved the Table.S2-3 to the main manuscript.

*I have the feeling the text also improved a lot (I believe the tracked changes version does not contain all changes), but think there are still some unclarities in the manuscript. For example, referencing the subfigures with labels from a-z (please see also the HESS guidelines for subfigures) and referring to the exact subfigures would probably help a lot already in discussing the different panels. See also my minor comments for more specific issues.*

We added subfigure lables on the Figure 9.

*I believe these comments are rather minor, and mainly textual. I hope the authors find them useful again, and look forward to a final version of the manuscript.*

We accept the suggestions for the technique corrections, hereafter we response the ones should be answered.

*P4.L119-120. Pixels classified...or snow free → and vice versa, correct?*

Not vice versa, only the missing data in Terra was replaced by the Aqua. See Parajka and Blöschl, 2008

*P8.L202. Why should SWE be bigger than 10mm?*

Parajka and Blöschl (2008) examined the sensitivity of SWE threshold for the snow objective function. We Added this reference after this sentence.

*P11.L256-257. The SSC ⋯ Wq <0.3. → You mean compared to the calibration or the other variant?*

It is compared to the other calibration variants.

*P13.L301. Are the soil moisture data used here actually the same as for the calibration? So remotely sensed soil moisture?*

Yes, they are the same as for the calibration.

*P20. This page is now only about supplementary material. Why not add one figure with the p-values?*

For highlighting the P-value which were less than 0.05, we prefer using the table. In response to this comment, we move back the Table S2-3 to the main manuscript.

**Response to referee #2**

*The authors have well addressed my comments on the technique and presentation issues indicated in the last round of review, and necessary changes have been made to improve the manuscript. One additional comment provided to the authors for consideration in the future: I think ASCAT soil moisture has still coarse spatial resolution that may not be that helpful for calibrating hydrological models in this type of analyses. In the future the authors may want to*

*consider high-spatial-resolution (tens of meters to 1 km) surface soil moisture generated using machine learning and multisource remote sensing data (e.g., Abowarda et al. 2021, RSE; Long et al. 2019, RSE), to see if the model performance on streamflow simulation can be further improved. I think this manuscript can now be accepted for publication in the prestigious journal of HESS.*

We want to thank the reviewer for her/his positive comments on the manuscript. And many thanks for the suggestions for our further study. Indeed, the finer spatial resolution soil moisture products have potential value in hydrological modelling, though our experiences from the ERS to ASCAT. In the future, we would have great interests to test the SSM generated using machine learning and multisource remote sensing data.

In response, we added a sentence in the discussion: "In the future studies, the use of soil moisture products with much finer spatial resolution may help reducing these errors and deficiencies for calibrating hydrological models (e.g. Bauer-Marschallinger et al., 2019; Long et al., 2019; Vergopolan et al., 2020; Abowarda et al., 2021).

Reference

Abowarda, A. S., Bai, L., Zhang, C., Long, D., Li, X., Huang, Q., and Sun, Z.: Generating surface soil moisture at 30 m spatial resolution using both data fusion and machine learning toward better water resources management at the field scale, Remote Sensing of Environment, 255, 112301, https://doi.org/10.1016/j.rse.2021.112301, 2021.

Bauer-Marschallinger, B., Freeman, V., Cao, S., Paulik, C., Schaufler, S., Stachl, T., Modanesi, S., Massari, C., Ciabatta, L., Brocca, L., and Wagner, W.: Toward Global Soil Moisture Monitoring With Sentinel-1: Harnessing Assets and Overcoming Obstacles, IEEE Transactions on Geoscience and Remote Sensing, 57, 520-539, 10.1109/TGRS.2018.2858004, 2019.

Long, D., Bai, L., Yan, L., Zhang, C., Yang, W., Lei, H., Quan, J., Meng, X., and Shi, C.: Generation of spatially complete and daily continuous surface soil moisture of high spatial resolution, Remote Sensing of Environment, 233, 111364, https://doi.org/10.1016/j.rse.2019.111364, 2019.

Vergopolan, N., Chaney, N. W., Beck, H. E., Pan, M., Sheffield, J., Chan, S., and Wood, E. F.: Combining hyper-resolution land surface modeling with SMAP brightness temperatures to obtain 30-m soil moisture estimates, Remote Sensing of Environment, 242, 111740, https://doi.org/10.1016/j.rse.2020.111740, 2020.